# PROMISSING: PRUNING MISSING VALUES IN NEURAL NETWORKS

## ABSTRACT

While data are the primary fuel for machine learning models, they often suffer from missing values, especially when collected in real-world scenarios. However, many off-the-shelf machine learning models, including artificial neural network models, are unable to handle these missing values directly. Therefore, extra data preprocessing and curation steps, such as data imputation, are inevitable before learning and prediction processes. In this study, we propose a simple and intuitive yet effective method for pruning missing values (PROMISSING) during learning and inference steps in neural networks. In this method, there is no need to remove or impute the missing values; instead, the missing values are treated as a new source of information (representing what we do not know). Our experiments on simulated data, several classification and regression benchmarks, and a multi-modal clinical dataset show that PROMISSING results in similar prediction performance compared to various imputation techniques. In addition, our experiments show models trained using PROMISSING techniques are becoming less decisive in their predictions when facing incomplete samples with many unknowns. This finding hopefully advances machine learning models from being pure predicting machines to more realistic thinkers that can also say "I do not know" when facing incomplete sources of information.

## 1 INTRODUCTION

Missing and incomplete data are abundant in real-world problems; however, the learning and inference procedures in machine learning (ML) models highly rely on high-quality and complete data. Therefore, it is necessary to develop new methods to deal with data imperfections in rugged environments. Currently, the most popular way to deal with imperfect data is to impute the missing values. However, if we consider the learning and inference procedures in our brain as a role model for ML algorithms, data imputation barely follows the natural principles of incomplete data processing in our brain. This is because the imputation is generally based on using a heuristic for *replacing* missing values. Our brain does not impute incomplete sensory information but instead uses its incompleteness as a separate source of information for decision making. For example, by only hearing the rain we can estimate how hard it is raining, and we do not necessarily need to receive visual information. Instead, we direct our attention more toward our auditory inputs to decide whether to go out with an umbrella. In addition, the more we miss sensory information, the more cautious we get in decision-making. That is why we are more careful in darker environments.

Neural networks (NNs) are brain-inspired algorithms that are very popular these days (under the name of deep learning) for learning complex relationships between inputs and target variables. However, they are in principle unable to handle incomplete data with missing values. They mainly rely on matrix operations which cannot operate on not-a-number (NaN) values. Only one NaN in a dataset impairs the forward propagation in a network. There are three solutions to this problem (García-Laencina et al., 2010): i) removing samples or features with missing values, ii) imputing the missing values, and iii) modeling the incomplete data. Removing the samples with missing values can be very costly, especially in small-sample size and high-dimensional datasets. For example, data collection in clinical applications is an expensive procedure in terms of time, finance, and patient burden. Moreover, removing even a few samples from small datasets can affect negatively the generalization performance of the final model. Removing informative features with missing values is also compensated with lower model performance. Therefore, filling the information gaps is inevitable.

There are various techniques for data imputation, ranging from simply imputing missing values with a constant to more sophisticated ML-based imputation approaches (Little & Rubin, 2019; García-Laencina et al., 2010). One can categorize the most common techniques into three main categories: i) constant imputation, ii) regression-based imputation, and iii) ML-based imputation. In constant imputation, the missing values are replaced with a constant, *e.g.*, zeros or mean/median of features. It has been shown that constant imputation is Bayes consistent when the missing features are not informative (Josse et al., 2019). In the regression-based imputation, a linear or non-linear regression model is derived to predict the missing values. This method can be used to impute a single or multiple features. The most popular regression-based imputation is Multiple Imputation by Chained Equations (MICE) (Van Buuren & Groothuis-Oudshoorn, 2011; Azur et al., 2011). In MICE, an iterative procedure of predicting missing values and re-training regressors with updated predictions is performed for a limited number of cycles. The central assumption behind the MICE approach is that the missing values are missed at random (see Rubin (1976) and Appendix A.1 for definitions of missing value mechanisms including missing completely at random (MCAR), missing at random (MAR), and missing not at random (MNAR)). Applying MICE can result in biased estimations if this assumption is not satisfied (Azur et al., 2011). Another critical limitation of regression-based imputation is its high computational complexity (Caiafa et al., 2021). If we do not know which feature will be missed at the test time, for $d$ features, we need to train $d$ different regression models. In the ML-based approach ML algorithms, such as a K-nearest neighbor (KNN), regularized linear model (Jiang et al., 2021), decision trees (Twala et al., 2008), random forest (Xia et al., 2017), neural network (Bengio & Gingras, 1996), or generative model (Yoon et al., 2018; Ipsen et al., 2020; Collier et al., 2020; Nazabal et al., 2020), are used for handling missing data.

As an alternative solution to data imputation, one can use the elegance of probabilistic modeling to model the incomplete data under certain assumptions. One seminal work in this direction is presented by Ghahramani & Jordan (1994), where a Gaussian Mixture Model (GMM) is used to estimate the joint density function on incomplete data using an Expectation-Maximization (EM) algorithm. This approach is later adopted and extended to logistic regression (Williams et al., 2005), Gaussian processes, support vector machines (Smola et al., 2005), and multi-class non-linear classification (Liao et al., 2007). However, despite their good performance on small-size datasets, their application remained limited on big and high-dimensional data due to the high computational complexity (Caiafa et al., 2021). To overcome this issue, Caiafa et al. (2021) proposed a sparse dictionary learning algorithm that is trained end-to-end, and simultaneously learns the parameters of the classifier and sparse dictionary representation. Le Morvan et al. (2020) proposed NeuMiss, a neural-network architecture that uses a differentiable imputation procedure in a impute-then-regress scheme (Morvan et al., 2021). A notable feature of NeuMiss is its robustness to MNAR data. Inverse probability weighted estimation (Wooldridge, 2007; Seaman & White, 2013) is another probabilistic approach for handling missing values without imputation in which the weights for samples with many missing values are inflated based on an estimation of the sampling probability. Recently, Smieja et al. (2018) proposed a modified neuron structure that uses GMM with a diagonal covariance matrix (assuming MAR) to estimate the density of missing data. GMM parameters are learned with other network parameters. Conveniently, it handles missing values in the first layer of the network, and the rest of the architecture remains unchanged. Elsewhere, Nowicki et al. (2016) proposed a new neural network architecture based on rough set theory (Pawlak, 1998) for learning from imperfect data. It is fascinating that this method can say "I do not know" when a large portion of input values are missing, unlike traditional models trained on imputed data that may predict definite outcomes even on completely unmeasured samples, *i.e.*, they run in the absolute darkness. These predictions can be dangerous with catastrophic consequences in more delicate applications of ML for example in autonomous driving, robotic surgery, or clinical decision-making.

In this work, we attack the problem of modeling incomplete data using artificial neural networks without data imputation. We propose a simple technique for pruning missing values (PROMISSING) in which the effect of missing values on the activation of a neuron is neutralized. In this strategy, a missing value is not replaced by arbitrary values (*e.g.,* through imputation); it is naturally considered a missing piece of the puzzle; *we learn a problem-specific numerical representation for unknowns*. The key feature of PROMISSING is its simplicity; it is plug-and-play; it deals with missing values in the first layer of the network without the need to change anything in the rest of the network architecture or optimization process. PROMISSING in its original form does not add extra parameters to the network, and its computational overhead remains negligible. Our experiments on simulated data and several classification/regression problems show that the proposed pruning

method does not negatively affect the model accuracy and provides competitive results compared to several data imputation techniques. In a clinical application, making prognostic predictions for patients with a psychotic disorder, we present an application of PROMISSING on a multi-modal clinical dataset. We demonstrate how the NN model trained using PROMISSING becomes indecisive when facing many unknowns. This is a crucial feature for developing trustworthy prediction models in clinical applications. Furthermore, we show a side application of PROMISSING for counterfactual interpretation (Mothilal et al., 2020) of NNs decisions that can be valuable in clinics.

## 2 METHODS

Let $x \in \mathbb{R}^p$ represent a vector of an input sample with $p$ features. We assume that the features in $x$ are divided into two sets of $q$ observed $x^o \in \mathbb{R}^q$ and $r$ missing features $x^m$ (where $p = q + r$). In this study, we do not put any assumption on the pattern of missing values in $x$. Then, the activation of the $k$th ($k \in \{1, 2, \ldots, s\}$) neuron in the first hidden layer of an ordinary NN is:

$$a^{(k)} = \sum_{x_i \in x^o} x_i w_i^{(k)} + \sum_{x_j \in x^m} x_j w_j^{(k)} + b^{(k)}. \tag{1}$$

This activation cannot be computed unless the values in $x^m$ are imputed with real numbers. Here, in PROMISSING, we propose to alternatively replace the missing values with a *neutralizer* that 1) prunes the missing values from inputs of a neuron, 2) neutralizes the effect of missing values on the neuron's activation by cancelling the second term in Eq. 1 and modifying the neuron's bias. A missing value $x_j \in x^m$ is replaced with its corresponding neutralizer $u_j^{(k)}$ at the $k$th neuron, where:

$$u_j^{(k)} = \frac{-b^{(k)}}{p w_j^{(k)}}. \tag{2}$$

The value of a neutralizer depends on its corresponding weight ($w_j^{(k)}$), the bias of the corresponding neuron ($b^{(k)}$), and the number of features ($p$); thus, it can be computed on the fly during the training or inference procedures. A small value is added to weights before computing neutralizers to avoid division by zero. Inserting the neutralizer into Eq. 1, the activation of the $k$th neuron is rewritten as:

$$a^{(k)} = \sum_{x_i \in x^o} x_i w_i^{(k)} + \frac{q b^{(k)}}{p}, \tag{3}$$

in which the effect of weights of missing values on the activation of the neuron is eliminated, and the neuron's bias is reduced by a factor of $r/p$. If all input values for a specific sample are missing then the neuron is completely neutralized.

**Proposition 1** *If all input values are missing ($q = 0$ and $x^o = \varnothing$) then the activation of a PROMISSING neuron is zero (see Appendix A.2.2 for the proof).*

**Proposition 2** *If there are no missing values in inputs ($q = p$ and $x^m = \varnothing$) then the activation of a PROMISSING neuron is equal to a normal neuron (see Appendix A.2.3 for the proof)..*

We should emphasize that, when using PROMISSING, the user does not need to apply any change to the input vectors, and the missing values (generally represented as nans in the input matrix) are fed directly to the network. After the training procedure, we eventually learn $U \in \mathbb{R}^{s \times p}$, a matrix representation for unknowns (or metaphorically the dark matter):

$$u_j^{*(k)} = \frac{-b^{*(k)}}{p w_j^{*(k)}} \qquad j \in \{1, 2, \ldots, p\}, \quad k \in \{1, 2, \ldots, s\}, \tag{4}$$

where $b^*$ and $w^*$ are representing the final learned bias and weight. At the prediction stage, a missing value at $j$th input feature will be replaced with its corresponding neutralizer from $U$. It is worth emphasizing that the missing values are replaced with different neutralizers at different neurons; therefore, it cannot be considered a constant imputation. In fact, each neuron perceives differently a missing value in the input space. Metaphorically, the neurons can be seen as blind men in the

parable of "the blind men and an elephant" [1] when facing unknowns. Furthermore, it is different from regression-based imputation and model-based approaches in the sense that a missing value in a specific feature is not inferred from other observed features, or the distribution of observed values; *i.e.*, unknowns remain unknowns. In PROMISSING, we do not assume any certain missing value mechanism (*e.g.*, MAR) in advance. Instead, we try to learn the patterns of missing values from data that maybe advantageous in more difficult scenarios such as MNAR (see results in Sec. 3.1).

One possible drawback of using PROMISSING is in high-dimensional input spaces and when the number of missing values is large, *i.e.*, when $p \to \infty$ and $r \gg q$. In this case, the neuron will undershoot; hence the effect of few non-missing values are ignored. [2] To address this problem, we propose a *modified* version of PROMISSING (mPROMISSING) in which the effect of large $r$ can be compensated with a *compensatory* weight, $w_c$. The compensatory weight receives a fixed input of $r/p$ for a specific sample; thus, the activation of the neuron will change to:

$$a^{(k)} = \sum_{x_i \in \boldsymbol{x}^o} x_i w_i^{(k)} + \frac{q b^{(k)} + r w_c^{(k)}}{p}. \tag{5}$$

$w_c^{(k)}$ is learned alongside the rest of the network parameters in the optimization process. On training data with few missing values, data augmentation (*e.g.*, by simulating different patterns and size of missing values) is advisable to ensure the sensibility of the learned compensatory weight. Since the input for this weight ($r/p$) is computed at the run time, no modification to input vectors is required.

**Proposition 3** *If there are no missing values ($q = p$, $r = 0$, and $\boldsymbol{x}^m = \varnothing$) then the activation of an mPROMISSING neuron is equal to a normal neuron (see Appendix A.2.4 for the proof).*

The proposed PROMISSING approach is straightforward to implement and use. It can be incorporated into the current implementations of different types of NN layers by adding/modifying few lines of code. We have implemented a `nanDense` layer, inheriting from Keras (Chollet et al., 2015) `Dense` layer, using PROMISSING and mPROMISSING neurons (see appendix A.3). The `nanDense` layer can be directly imported and used with any Keras model. In its general usage, the `nanDense` layer is only used for the first layer of an NN to handle missing values in inputs, unless we expect some missing values in the intermediate layers.

## 3 EXPERIMENTS AND RESULTS

In a set of three experiments, 1) in a simulation study, we investigate the convergence behaviors of (m)PROMISSING on MCAR, MAR, and MNAR data; 2) in a large experiment on real data, we benchmark the reliability of (m)PROMISSING in several classification and regression tasks, and 3) on a multi-modal clinical dataset, we demonstrate an application of PROMISSING on an open problem of psychosis prognosis prediction.

### 3.1 SIMULATED DATA

We conducted a simple analysis of synthesized data to better understand the behaviors of PROMISSING. For data simulation, we simulated the XOR problem with 1000 samples. Gaussian noise with a variance of 0.25 is added to data. We simulated the missing data for MCAR, MAR, and MNAR settings. The same procedures as in Schelter et al. (2021) are used to simulate the MCAR, MAR, and MNAR missing values (see Appendix A.4 for details). The experiments are repeated for 30% and 50% missing samples. We used scikit-learn (Pedregosa et al., 2011) for implementing different data imputation schemes.[3] We also compared our method with a GAN-based imputation proposed in Yoon et al. (2018). Default configurations are used for all imputers.

To evaluate (m)PROMISSING, we used a simple fully connected NN architecture with four tangent-hyperbolic neurons in the hidden layer and a single sigmoid neuron in the output layer. This simple architecture can reach the performance of the Bayes optimal classifier (AUC=$0.99 \pm 0.01$) on simulated data without missing values (see the black lines in Fig. 1). We randomly divided the simulated

---

[1]See `https://en.wikipedia.org/wiki/Blind_men_and_an_elephant` for the parable.
[2]Note that this again might be considered as an advantage in some applications.
[3]Scikit-learn 0.24.2 with Numpy 1.19.5.

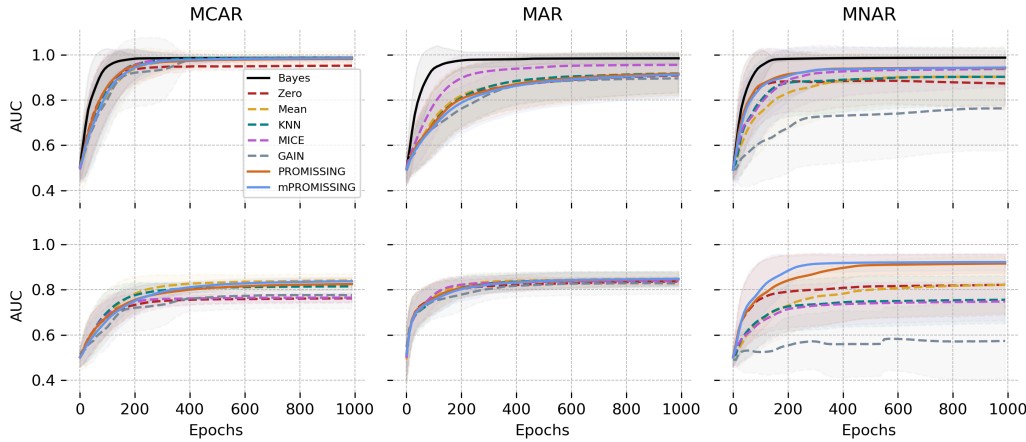

Figure 1: Comparison between learning curves of PROMISSING and mPROMISSING with data imputation approaches on simulated MCAR, MAR, and MNAR data, when tested on a test set without missing values (first row) and with $50\%$ missing values (second row).

data into training and test sets with 500 samples in each set. The pipeline of 1) data simulation, 2) random data corruption with missing values, 3) randomly splitting data into the training and test set, and 4) training the model is repeated 100 times, and the standard deviation of performances across repetitions is reported (shadowed areas in Fig. 1). We have fixed the random number generator seed in each repetition (randomly selected from $[0, 100000]$) to ensure a fair comparison between models. A stochastic gradient descent (SGD) optimizer is used to minimize a binary cross-entropy loss function. We evaluated the network performance as the training procedure proceeds in two different settings i) on the complete test set without missing data, and ii) on the test set with missing values.

The first row of plots in Fig. 1 shows the area under the ROC curve (AUC) for models trained on data with $50\%$ missing values (see Fig. 7 for $30\%$) and tested on test sets without missing values as the number of epochs progresses. Here, by comparing the performance gap between the different missing data curation strategies and the optimal Bayes classifier, we can evaluate the *training bias* imposed by each strategy during the training. A lower training bias indicates a lower effect of data curation strategy on the quality of the trained model. Our empirical results show that the training bias is modulated by both the strategy and missing data mechanism. In the MCAR setting, all strategies show less training bias. However, in the MAR and MNAR conditions, the model biases are more pronounced, especially with higher percentages of missing values. The (m)PROMISSING techniques show a competitive performance, especially in the MNAR setting where they perform equivalently with mean, KNN, and MICE imputers (with Wilcoxon rank-sum test p-values of 0.26, 0.02, and 0.15) and better performance than zero and GAIN imputations (p-values$\leq 3 \times 10^{-12}$). These results confirm that the (m)PROMISSING models provide the possibility to learn meaningful representations for the unknowns that are at least as informative as imputed data.

The second row of plots in Fig. 1 compares the learning curves of models when tested on the test set with missing values. By comparing the final performance of models in the first and second row, we immediately notice the performance drops across almost all strategies. These performance drops show the negative effect of data degradation imposed by each imputation method on the generalization of models when applied to data with missing values. Thus, we refer to this difference as *test bias*. While all methods show similar test bias in the MCAR and MAR scenarios, the (m)PROMISSING models show a negligible test bias in the MNAR scenario, *i.e.*, the performance of (m)PROMISSING models is not affected when they are applied to test data with MNAR missing values. Furthermore, m(PROMISSING) methods show significantly better performance on MNAR test data (p-values$\leq 6 \times 10^{-6}$). The small training and test biases of the (m)PROMISSING models in the MNAR setting confirm their effectiveness in jointly modeling the data and missing patterns.

We have further analyzed the patterns of neutralizers in $\boldsymbol{U}$ to interpret the learned representations of unknowns in PROMISSING (see Fig. 8 in appendix A.5). Since $\boldsymbol{U}$ is derived from the network parameters, due to the stochastic nature of parameter initialization and optimization, it may end

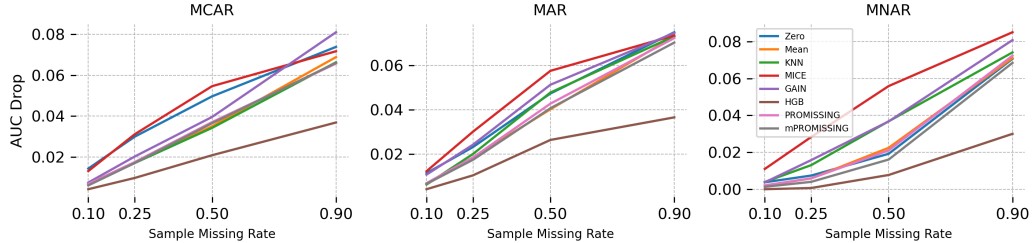

Figure 2: Comparing AUC drops in classification tasks with respect to the full model across five different imputation methods, HGB, and (m)PROMISSING in MCAR, MAR, and MNAR settings.

up with different values in each repetition. Our visual inspection shows that each neuron learns a different solution for unknowns, ranging from a simple zero or mean imputer to more complex patterns. This is a unique feature for PROMISSING providing the possibility to learn a neuron-specific imputation strategy from data.

## 3.2 OPENML DATA

In the second set of experiments, we benchmarked (m)PROMISSING alongside several imputation techniques on a range of publicly available classification and regression datasets from OpenML (Vanschoren et al., 2014). We used the same classification and regression datasets as in Jäger et al. (2021) (see Table 2 and Table 3 in appendix A.6). Here, we use a simple fully-connected architecture with two hidden layers, with $p/2$ ReLU neurons in the first and two neurons in the second hidden layer, so the architecture changes slightly from one dataset to another depending on the number of features. A single output neuron with a sigmoid or linear transfer function is used for output layers in classifiers or regressors, respectively. Our quick experiments on benchmark datasets show that this simple architecture provides competitive results with random forest classifiers and regressors (see Fig 10 in appendix A.6). We excluded three classification and three regression datasets from our analyses due to the poor performance of baseline models on the complete data (see appendix A.6). We used MCAR, MAR, and MNAR schemes to simulate different ratios (0.1, 0.25, 0.5, 0.9) of samples with missing values across datasets. The same procedures as in Schelter et al. (2021) are used to simulate the MAR and MNAR missing values. In the MNAR setting, a random percentile of the most informative feature (based on the mutual information with the target variable) is removed. In the MAR case, values in the most informative feature are removed based on a random percentile of the second most informative feature (see Appendix A.4 for more details). An SGD optimizer for 100 epochs with a mini-batch size of 10 samples is used in the training process to minimize a binary cross-entropy or a mean squared error (MSE) loss in the classification or regression scenario. The performances of different methods are evaluated in a 2-fold cross-validation scheme computing the AUC and standardized MSE (SMSE) metrics. The whole experiment pipeline, including 1) simulating missing values, 2) data imputation (this step does not apply to (m)PROMISSING), 3) data standardization, 4) model training and evaluation, is repeated twenty times with a fixed random number generator seed in each repetition. When imputing data, the default imputer settings are applied. We have also included the histogram-based gradient boosting (HGB) (Ke et al., 2017) –that similar to (m)PROMISSING provides a mechanism to directly deal with missing values without imputation– in our experiments.

Fig. 2 compares the performance of different data imputation methods with (m)PROMISSING in classification tasks for various missing value mechanisms and missing sample ratios (x-axes). The AUC drops with respect to the full model (trained and tested on complete data) are plotted. To summarize the results across datasets and replication runs, we first compute the median performance across datasets and then average across twenty replications. While in all settings the HGB is the best performer, the (m)PROMISSING shows competitive performance with the better performing imputers. A similar overall pattern can be observed when comparing the increase in the SMSE in regression tasks (see Fig. 11 in appendix A.6). In summary, these results show that (m)PROMISSING can compete with data imputation approaches without the need for filling the unknowns, thus it is more flexible with underlying missing data mechanisms and all input data types including continuous, binary, and categorical.

### 3.3 Clinical Application: Psychosis Prognosis Prediction

Missing data are seemingly ubiquitous in the healthcare domain. In general, clinical datasets contain multi-modal data that are collected through different measurement tools, including patient questionnaires, medical reports and instruments, biological tests, and imaging data. Each modality may represent a different aspect of a disease or treatment outcome. Therefore, employing smart modality fusion techniques is inevitable to best benefit from these rich data. However, this goal is occluded because these data are collected through different sources and often include missing variables or even missing modalities. In this section, we demonstrate an application of PROMISSING in multi-modal data fusion and counterfactual model interpretation for psychosis prognosis prediction (PPP).

#### 3.3.1 OPTiMiSE Dataset

We used the OPTiMiSE dataset (Kahn et al., 2018), an antipsychotic three-phase switching study: 495 patients with schizophrenia, schizophreniform, or schizoaffective disorder according to DSM-IV diagnosis standards. The patients received different medications in a three-phase study design. We used the data from the first phase of the study in which the patients were treated with amisulpride for four weeks. Only 376

Table 1: Data modalities in the OPTiMiSE dataset.

| | Modality | Measures Num. | Samples with NaNs | Percentage of NaNs | Description |
|---|---|---|---|---|---|
| 1 | Demographics | 20 | 300 | 16% | Socio-demographic features |
| 2 | Diagnosis | 6 | 45 | 6% | Illness related features |
| 3 | Lifestyle | 7 | 421 | 21% | Use of substances like drugs and alcohol |
| 4 | Somatic | 11 | 92 | 13% | Physical examination |
| 5 | Treatment | 1 | 67 | 14% | Average dosage of medication |
| 6 | MINI | 67 | 47 | 8% | Psychiatric comorbidity |
| 7 | Cytokines | 34 | 100 | 20% | Proteins important in cell signaling |
| 8 | PANSS | 30 | 42 | 8% | Positive and Negative Syndrome Scale |
| 9 | PSP | 5 | 57 | 11% | Personal and Social Performance Scale |
| 10 | CGI | 1 | 46 | 9% | Clinical Global Impression |
| 11 | CDSS | 9 | 53 | 11% | Measurement scale about depression |
| 12 | SWN | 20 | 81 | 16% | Subjective Well-Being Under Neuroleptic |
| | Summary | 211 | - | 13% | - |

patients who finished the first phase are used in our experiments. In a PPP framework, we aimed to predict the probability of patients' symptomatic remission based on the Positive and Negative Syndrome Scale (PANSS) as defined by consensus criteria (Andreasen et al., 2005). We used 12 different data modalities in our experiments (see Table 1). In total, 13% values in the dataset are missing. These missing values include complete or partial missed modalities. The missing values are scattered across patients and measures; somehow, if we remove all subjects with missing values only 17 patients are left. This observation substantiates, even more, the importance of missing data management in clinical datasets. We have applied a minimal preprocessing pipeline on the data before the modeling stage. The input data contains binary, categorical, and continuous variables. We used 0/1 and one-hot encoding for binary and categorical variables, respectively. All continuous variables are standardized before feeding them to the model. Considering the unbalanced class distributions (67% of patients get remitted), in each fold of K-fold cross-validation, we resampled the minority class samples in the training data to match its size to the majority class.

#### 3.3.2 A Multi-Modal Model for PPP

Inspired by Huang et al. (2020), we used a multi-modal NN architecture with middle and late information fusion. As depicted in Fig. 3, our model has five layers: input, representation learning, modality-specific classification, fusion, and classification. The network receives the data from $M$ different modalities in the input layer. Since input samples may contain missing values, we use `nanDense` layers right after the input layer in the representation learning phase. These layers transfer the raw inputs to an intermediate representations. We roughly (without any optimization and solely based on input feature size) fixed the number of neurons in this layer for each modality (Demographics=10, Diagnosis=10, Lifestyle=5, Somatic=5, Treatment=2, MINI=10, Cytokines=10, PANSS=10, PSP=5, CGI=2, CDSS=5, SWN=5). In the modality-specific classification layer, we use 2 Softmax neurons

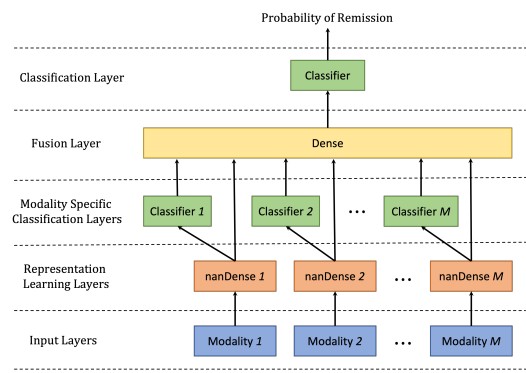

Figure 3: The NN architecture with middle and late information fusion for psychosis prognosis prediction on multi-modal data.

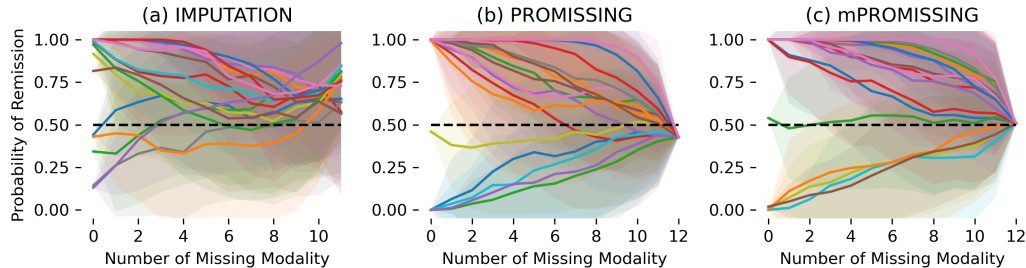

Figure 4: The trajectories of model predictions for 17 patients with information loss when using a) data imputation, b) PROMISSING, and c) mRPOMISSING methods to handle missing data.

to classify the modality-specific representations into target classes, *i.e.*, whether a patient remits given a certain amount of medication after four weeks. These decisions are merged with middle representation and then fed to the fusion layer. A trivial ReLU `Dense` layer with five neurons is used for the fusion layer. Finally, two Softmax neurons are used to classify the fused data into target classes. Dropout layers with a probability of $0.1$ are applied before any weighted layer. An Adam (Kingma & Ba, 2014) optimizer (with learning rate of $0.0003$) is used to minimize the total categorical cross-entropy loss across classification layers.

### 3.3.3 EVALUATION AND COMPARISON

We first compared (m)PROMISSING with two alternative strategies: i) removing features with missing values and ii) data imputation. In the first case, we first removed subjects that completely missed one data modality; then, we removed features containing missing values. This procedure ended up losing 67 subjects and $78\%$ of features including two modalities. In the second case, we used a KNN to impute the missing data. Note that, due to binary and categorical features, many imputation techniques (such as mean and iterative imputer) cannot be directly applied to our data. Even in the KNN case, we needed to set the number of neighbors equal to one to avoid float numbers for the categorical features. A 10-fold cross-validation scheme is employed for model evaluations, and all the experiments are repeated ten times to evaluate the variability in performances. The results support our conclusion in Sec. 3.2 that (m)PROMISSING performs as well as data imputation. PROMISSING and mPROMISSING result in AUCs of $0.67 \pm 0.02$ and $0.66 \pm 0.02$ that are on a par with the KNN imputer $0.66 \pm 0.02$. However, as is expected, removing missing features resulted in an impaired model performance of $0.61 \pm 0.02$. The similar performance of (m)PROMISSING and data imputation despite their completely different strategies to deal with missing values raises two questions: "Do the resulting models behave similarly too?", "If not, how are they different?". Craving to answer these questions, we set up the second set of experiments on the OPTiMiSe data.

In the next experiment, we left all the 17 patients with complete data in the test set and used the other 359 patients (with missing data) for training. Here, to fully use the capabilities of (m)PROMISSING models, we have augmented the training data by removing different subsets of modalities from data. Having $M = 12$ modalities and $n = 470$ samples (after balancing the training data) in the training set, this procedure resulted in $1,919,010$ samples ($\sum_{m=0}^{M-1} \binom{M}{m} \times n$). We trained three models, one using KNN imputation and two using (m)PROMISSING. Then, we have used a particular procedure for testing these models on the test set. In 100 repetitions, we have removed the modalities one by one and in random order. Fig. 4 shows the trajectories of predictions for 17 test subjects across three models. The trajectories show the predicted probability of remission when the model has access to less information about the patients (from complete information on the left to no information on the right). Note that it is impossible to remove all modalities in the KNN imputation case because it impairs the imputation mechanism. The interesting observation is the difference between the prediction behavior of models when they have less information available. The model trained and tested on imputed data always reacts to inputs even in an unknown environment. While the predictions of m(PROMISSING) models monotonically converge to 0.5, *i.e.*, the models become indecisive in an unknown environment. As expected, the predictions of the PROMISSING model are slightly below the chance when the number of missing values increases, but this divergence is corrected in mPROMISSING. This feature is crucial in delicate applications of ML in the medical domain in which the decisions made by machines can significantly affect the quality of life of patients.

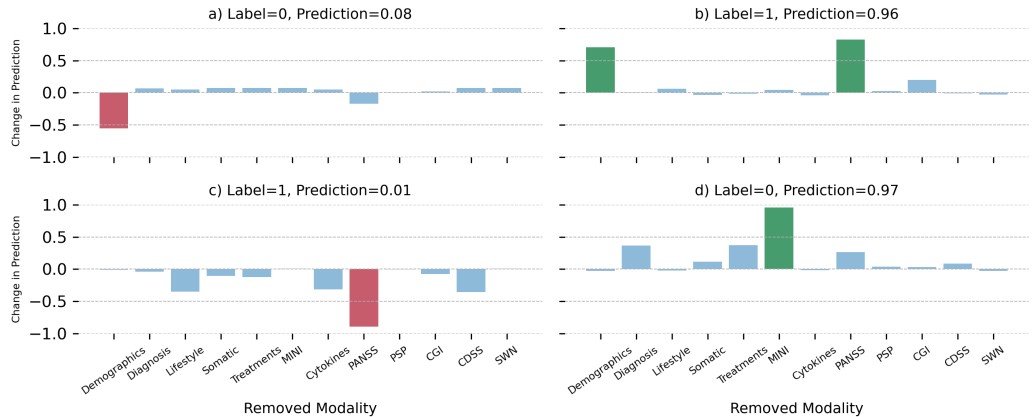

Figure 5: Counterfactual interpretation of model decisions using mPROMISSING for a) true negative, b) true positive, c) false negative, and d) false positive examples in model predictions.

One possible application of PROMISSING is in counterfactual interpretation (CI) (Mothilal et al., 2020) of model decisions. CI is an effective method for the causal interpretation of complex models such as NNs. PROMISSING facilitates the usage of CI; we can simply use artificially inserted missing values to create the counterfactual examples. In other words, to evaluate the effect of one variable or modality on the final prediction, we assume it is missing in a counterfactual scenario. To demonstrate this feature, we used this technique on the predictions of an mPROMISSING model. The first and second row of Fig. 5 show respectively the CI results for four example patients with correct and incorrect predictions on their remission. In the first case (Fig. 5a), the model correctly predicts no-remission, and CI shows that the demographic features are the decisive factor by reducing the probability of remission by $\sim 0.5$. Similar inference can be used to explain the false decisions. For example, PANSS and MINI measures are decisive factors in wrong model predictions in Fig. 5c and 5d, respectively. This extra level of model explainability at the individual patient level is valuable in the clinical settings as it answers the "Why?" question often asked by the clinicians about decisions of black box models, and hopefully, paves the way towards the new field of precision psychiatry (Fernandes et al., 2017).

## 4 SUMMARY AND CONCLUSIONS

In this work, we presented PROMISSING, a promising and novel *second* thought on dealing with missing data in neural networks. Unlike previous works, we do not intend to replace missing values based on their relation to other features or the joint distribution of observed data. Instead, we opt to use the unknowns as an additional source of information by learning a representation for missing data. The learned representation is used to prune missing values by neutralizing their effect on the neurons' activation. The proposed method is convenient, intuitive, and reliable. It is convenient because it is straightforward in implementation and application; there is no need for extra data preparation, and the missing values are directly fed to the model; and unlike some data imputation methods, it is flexible with all input data types including continuous, binary, and categorical. This feature makes PROMISSING irreplaceable in applications of neural networks on multi-modal datasets with a mixture of diverse features. It is intuitive because it reacts to unknowns like a living agent; the more the unknowns, the less the actions. Moreover, it is reliable since it performs as well as its alternatives while its decisions are calibrated with the amount of information it receives. Furthermore, it provides an embedded mechanism to explain the decisions of complex neural networks models which is favorable for more reliable applications of ML in clinical decision making. Therefore, we consider PROMISSING as one step ahead towards more naturalistic ML-based decision-making in uncertain environments. Despite our extensive experiments, some analytical and empirical aspects of PROMISSING remain unexplored. Exploring the connection between modeling unknowns and model uncertainty, empirical comparison with model-based incomplete data modeling, and applications on more complex and structured data are among the unexplored avenues that we consider as future work.

REPRODUCIBILITY STATEMENT

To ensure the reproducibility of conclusions of this work, we have made the core implementation of the proposed method available in appendix A.3. The `nanDense` layer can be directly imported and tested by the readers. Furthermore, anonymized scripts and codes to reproduce the results for experiments on simulated data and OpenML datasets are included in the supplementary materials. Meanwhile, in case needed, we are open to sharing scripts of the experiments on clinical data with reviewers during the discussion process (the data is not available publicly). In our largest validation experiments in Sec. 3.2, we have used 52 publicly available ML datasets from OpenML. We tried our best to be very specific, descriptive, and precise about the applied parameter settings, model specifications, and model evaluations in Sec. 3. We plan to release all the codes and scripts in Github upon acceptance of the manuscript.

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

# A APPENDIX

## A.1 MISSING DATA MECHANISMS

Let $X \in \mathbb{R}^{n \times p}$ represent the data matrix of $n$ samples with $p$ features. Each sample $x \in X$ then can be decomposed into observed $x^o$ and missing $x^m$ components where the pattern of the missing values can be indicated by a mask vector $m \in \{0, 1\}^p$. Using these notation, the underlying mechanisms for missing values in data can be divided into three categories (Rubin, 1976; Ghahramani & Jordan, 1995; Schafer & Graham, 2002):

1. **Missing Completely at Random (MCAR):** where there is no systematic relationship between the patterns of missingness with either observed or unobserved variables, *i.e.*, $p(m \mid x^o, x^m) = p(m)$. In this case the missingness patterns are completely at random and do not depend on observed or missing data.

2. **Missing at Random (MAR):** is a weaker assumption than MCAR (MCAR data is MAR but not vice-versa), in which the patterns of missingness are independent from missing data but may depend on observed values, *i.e.*, $p(m \mid x^o, x^m) = p(m \mid x^o)$.

3. **Missing Not at Random (MNAR):** where the pattern of missingness depends on the values of missing data, *i.e.*, $p(m \mid x^o, x^m)$ depends on the values in $x$. An example of MNAR is censored data in which a certain range of data is missing.

## A.2 ACTIVATION OF (M)PROMISSING NEURONS

### A.2.1 DERIVATION OF EQ. 3

Eq. 3 can be derived by inserting Eq. 2 in Eq. 1 for missing inputs $x_j \in x^m$:

$$a^{(k)} = \sum_{x_i \in x^o} x_i w_i^{(k)} + \sum_{x_j \in x^m} x_j w_j^{(k)} + b^{(k)} \quad \xrightarrow{x_j = \frac{-b^{(k)}}{p w_j^{(k)}}}$$

$$a^{(k)} = \sum_{x_i \in x^o} x_i w_i^{(k)} + \sum_{x_j \in x^m} \frac{-b^{(k)} w_j^{(k)}}{p w_j^{(k)}} + b^{(k)}$$

$$= \sum_{x_i \in x^o} x_i w_i^{(k)} + \frac{-r b^{(k)} + p b^{(k)}}{p} = \sum_{x_i \in x^o} x_i w_i^{(k)} + \frac{(p - r) b^{(k)}}{p}$$

$$= \sum_{x_i \in x^o} x_i w_i^{(k)} + \frac{q b^{(k)}}{p}.$$

### A.2.2 PROOF OF PROPOSITION 1

When all input values are missing ($x^o = \varnothing$, $q = 0$, and $r = p$), then the activation of a PROMISSING neuron is zero:

$$a^{(k)} = \sum_{x_i \in x^o} x_i w_i^{(k)} + \sum_{x_j \in x^m} x_j w_j^{(k)} + b^{(k)} \quad \xrightarrow[x^o = \varnothing]{x_j = \frac{-b^{(k)}}{p w_j^{(k)}}}$$

$$a^{(k)} = \sum_{x_j \in x^m} \frac{-b^{(k)} w_j^{(k)}}{p w_j^{(k)}} + b^{(k)} = \frac{-r b^{(k)}}{p} + b^{(k)} \quad \xrightarrow{r = p}$$

$$a^{(k)} = 0.$$

### A.2.3 PROOF OF PROPOSITION 2

When there are no missing values in inputs ($\boldsymbol{x}^m = \varnothing$, $q = p$, and $r = 0$), then the activation of a normal neuron (Eq. 1) is computed as follows:

$$a^{(k)} = \sum_{x_i \in \boldsymbol{x}^o} x_i w_i^{(k)} + \sum_{x_j \in \boldsymbol{x}^m} x_j w_j^{(k)} + b^{(k)} \quad \xrightarrow{\boldsymbol{x}^m = \varnothing}$$

$$a^{(k)} = \sum_{x_i \in \boldsymbol{x}^o} x_i w_i^{(k)} + b^{(k)}.$$

In this case, the activation of a PROMISSING neuron is exactly equal to an ordinary neuron:

$$a^{(k)} = \sum_{x_i \in \boldsymbol{x}^o} x_i w_i^{(k)} + \frac{q b^{(k)}}{p} \quad \xrightarrow{p=q} \quad a^{(k)} = \sum_{x_i \in \boldsymbol{x}^o} x_i w_i^{(k)} + b^{(k)}.$$

### A.2.4 PROOF OF PROPOSITION 3

When there are no missing values in inputs ($\boldsymbol{x}^m = \varnothing$, $q = p$, and $r = 0$), the activation of an mPROMISSING neuron is exactly equal to an ordinary neuron:

$$a^{(k)} = \sum_{x_i \in \boldsymbol{x}^o} x_i w_i^{(k)} + \frac{q b^{(k)} + r w_c^{(k)}}{p} \quad \xrightarrow[r=0]{p=q} \quad a^{(k)} = \sum_{x_i \in \boldsymbol{x}^o} x_i w_i^{(k)} + b^{(k)}.$$

### A.3 A NANDENSE LAYER

A preliminary implementation of a `nanDense` layer is listed in Listing 1. This layer is implemented by inheriting and overloading the class constructor, `build`, and `call` functions. A flag, `use_c`, is added to the class constructor to decide whether to use compensatory weights or not (defaulted to `False`). Using this flag, the user can simply switch between PROMISSING and mPROMISSING. Furthermore, the flag `use_bias` is fixed to `True`. In the `build` function an extra weight is concatenated to the end of the weight vector to implement compensatory weight in the case `use_c==True`. The matrix of epsilons is also pre-set for usage in the `call` function. In the `call` function, the values for compensatory weights and neutralizers are computed, accordingly. The current implementation is limited to non-sparse 2-dimensional inputs. The implementation is preliminary and not optimized for memory and speed efficiencies. This implementation is tested with `Python 3.8.3`, `TensorFlow 2.4.1`, and `Keras 2.4.3`. A similar guideline may be followed to implement PROMISSING within other types of NN layers.

```python
import keras
import tensorflow as tf
from tensorflow.python.framework import sparse_tensor
from tensorflow.python.ops import gen_math_ops
from tensorflow.python.ops import math_ops
from tensorflow.python.ops import nn_ops
from tensorflow.python.keras import backend as K
from tensorflow.python.framework import dtypes
from tensorflow.python.framework import tensor_shape
from tensorflow.python.keras.engine.input_spec import InputSpec
from tensorflow.python.keras import activations
from tensorflow.python.keras import initializers
from tensorflow.python.keras import regularizers
from tensorflow.python.keras import constraints

class nanDense(keras.layers.Dense):

    def __init__(self,
                 units,
                 use_c = False, # A flag to use compensatory weight or
    not.
                 activation=None,
```

```
23                kernel_initializer='glorot_uniform',
24                bias_initializer='zeros',
25                kernel_regularizer=None,
26                bias_regularizer=None,
27                activity_regularizer=None,
28                kernel_constraint=None,
29                bias_constraint=None,
30                **kwargs):
31        super(nanDense, self).__init__(units,
32                activity_regularizer=activity_regularizer, **kwargs)
33
34        self.use_c = use_c
35        self.use_bias = True
36        self.units = int(units) if not isinstance(units, int) else units
37        self.activation = activations.get(activation)
38        self.kernel_initializer = initializers.get(kernel_initializer)
39        self.bias_initializer = initializers.get(bias_initializer)
40        self.kernel_regularizer = regularizers.get(kernel_regularizer)
41        self.bias_regularizer = regularizers.get(bias_regularizer)
42        self.kernel_constraint = constraints.get(kernel_constraint)
43        self.bias_constraint = constraints.get(bias_constraint)
44        self.input_spec = InputSpec(min_ndim=2)
45        self.supports_masking = True
46
47    def build(self, input_shape):
48
49        dtype = dtypes.as_dtype(self.dtype or K.floatx())
50        if not (dtype.is_floating or dtype.is_complex):
51          raise TypeError('Unable to build 'nanDense' layer with non-
    floating point '
52                          'dtype %s' % (dtype,))
53
54        input_shape = tensor_shape.TensorShape(input_shape)
55        last_dim = tensor_shape.dimension_value(input_shape[-1])
56        if last_dim is None:
57            raise ValueError('The last dimension of the inputs to '
    nanDense' '
58                          'should be defined. Found 'None'.')
59        self.input_spec = InputSpec(min_ndim=2, axes={-1: last_dim})
60
61        if self.use_c:       # an extra weight if use_c is True.
62            self.kernel = self.add_weight(
63                'kernel',
64                shape=[last_dim+1, self.units],
65                initializer=self.kernel_initializer,
66                regularizer=self.kernel_regularizer,
67                constraint=self.kernel_constraint,
68                dtype=self.dtype,
69                trainable=True)
70        else:
71            self.kernel = self.add_weight(
72                'kernel',
73                shape=[last_dim, self.units],
74                initializer=self.kernel_initializer,
75                regularizer=self.kernel_regularizer,
76                constraint=self.kernel_constraint,
77                dtype=self.dtype,
78                trainable=True)
79
80        self.bias = self.add_weight(
81            'bias',
82            shape=[self.units,],
83            initializer=self.bias_initializer,
84            regularizer=self.bias_regularizer,
85            constraint=self.bias_constraint,
```

```
86              dtype=self.dtype,
87              trainable=True)
88
89        self.epsilon = tf.fill(self.kernel.shape, K.epsilon()) # Epsilon
     Matrix
90
91        self.built = True
92
93
94    def call(self, inputs):
95
96        dtype = self._compute_dtype_object
97
98        if self.use_c:  # Computing and concatenating the compensatory
     weight to the weights
99            c = tf.math.reduce_sum(tf.cast(tf.math.is_nan(inputs), dtype)
     , axis=1)
100           c = tf.math.divide(c, inputs.shape[1])
101           inputs = tf.concat([inputs,tf.expand_dims(c, axis=1)], axis
     =1)
102
103       kernel = math_ops.add(self.epsilon, self.kernel) # Adding epsilon
      to weights
104
105       if self.dtype:
106           if inputs.dtype.base_dtype != dtype.base_dtype:
107             inputs = math_ops.cast(inputs, dtype=dtype)
108
109       rank = inputs.shape.rank
110       if rank == 2 or rank is None:
111           if isinstance(inputs, sparse_tensor.SparseTensor):
112               raise NotImplementedError
113           else:
114               outputs = []
115               for i in range(self.kernel.shape[1]): # Computing
     Neutralizers and activations for each neuron
116                   d = tf.math.divide(-self.bias[i]/self.kernel.shape
     [0], kernel[:,i])
117                   temp_inputs = tf.where(tf.math.is_nan(inputs), d,
     inputs) # replacing nans in inputs with -b/w
118                   outputs.append(gen_math_ops.mat_mul(temp_inputs,
     kernel[:,i:i+1]))
119               outputs = tf.concat(outputs, axis=1)
120
121       # Broadcast kernel to inputs.
122       else:
123           raise NotImplementedError
124
125       outputs = nn_ops.bias_add(outputs, self.bias)
126
127       if self.activation is not None:
128           outputs = self.activation(outputs)
129
130       return outputs
```

Listing 1: An implementation of a `nanDense` layer in Keras.

### A.4 MISSING DATA SIMULATIONS

We used the same procedures in Schelter et al. (2021) to simulate the missing values in our experiments on simulated and OpenML data. For MAR and MNAR settings, first, a random range of a certain feature $F1$ is selected. This random range is decided based on random lower and upper bound percentiles. In the MAR condition, the values of a second random feature $F2$ are removed if the values of $F1$ lie in the specified range, *i.e.*, the missing values in $F2$ depend on the observations

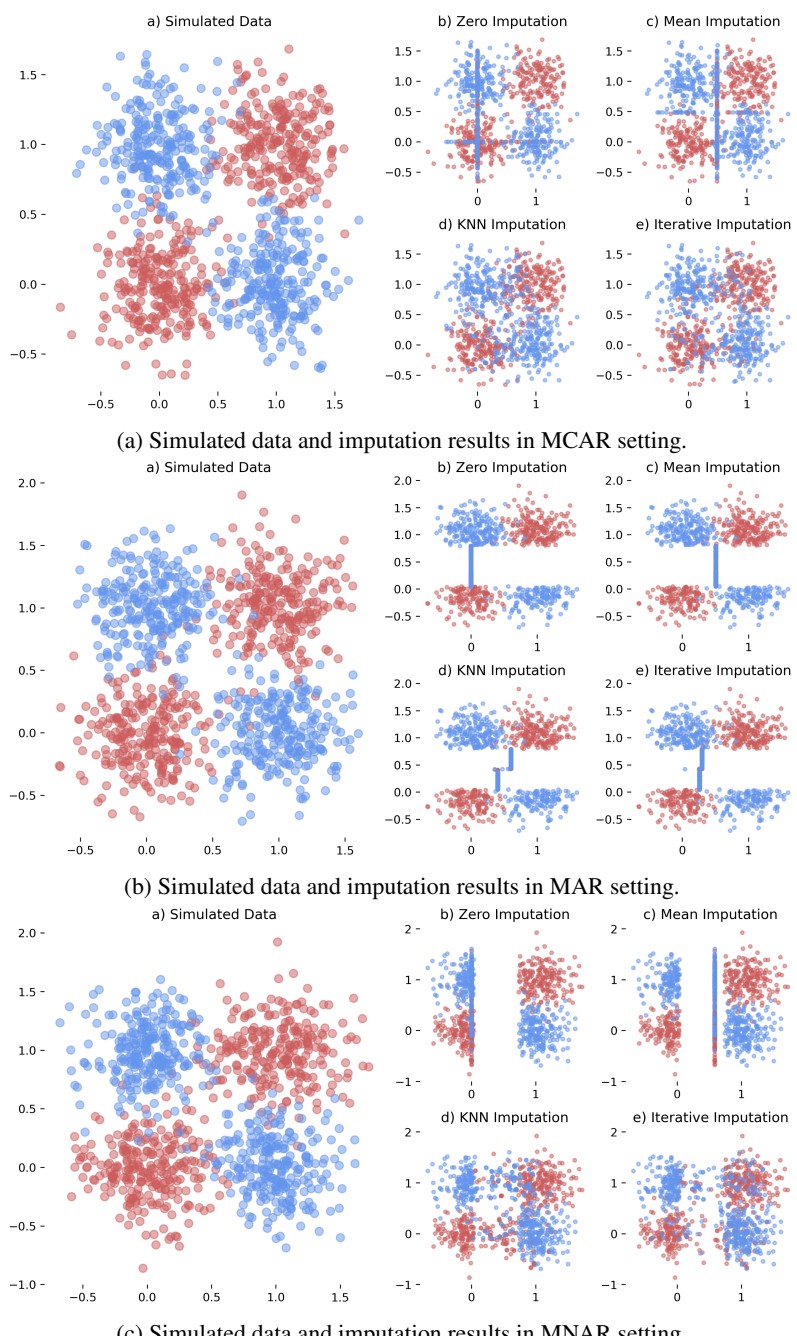

(a) Simulated data and imputation results in MCAR setting.

(b) Simulated data and imputation results in MAR setting.

(c) Simulated data and imputation results in MNAR setting.

Figure 6: In each panel: a) The simulated XOR data. The effect of b) zero, c) mean, d) KNN, and e) iterative imputations on simulated data with MCAR, MAR, and MNAR missing values. The missing values for the first feature (x-axis) are selected based on a random range of the second feature (y-axis).

in $F1$ (see Fig. 6b for an example). In contrast in the MNAR case, the values are removed from the same $F1$ feature. In fact, in the MNAR case the values of $F1$ are censored for in a certain range (see Fig. 6c for an example). In the MCAR setting, the samples with missing value are selected completely at random (see Fig. 6a).

Fig. 6b-e illustrate the effect of applying different data imputation methods. In all cases, imputation may result in the misrepresentation of samples with missing values.

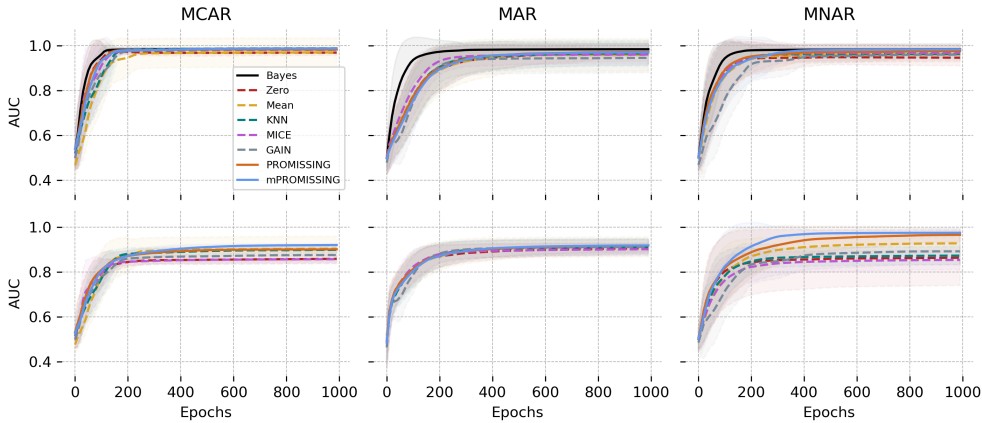

Figure 7: Comparison between learning curves of PROMISSING and mPROMISSING with data imputation approaches on simulated MCAR, MAR, and MNAR data, when tested on a test set without missing values (first row) and with 30% missing values (second row).

## A.5 SUPPLEMENTARY MATERIAL FOR THE SIMULATION STUDY

To better understand the learned representation of unknowns $U$, we have visualized the elements in $U$ across 100 simulation replications. Fig. 8 shows the values of $U$ across 100 simulation runs in our simulation study. In our simulation study, $U$ has eight values (two inputs for each four input neurons). Markers with similar color and shape represent the neutralizer values for two features (axes) in four neurons in a specific run. For example, the four blue circles show the values of four neutralizers in two-dimensional feature space in the first run of the experiment. Since the scatter plot is very crowded, we plotted only the 8 representative runs in Fig. 9 in the main text in Sec. 3.1. Since $U$ is derived from the network parameters, due to the stochastic nature of parameter initialization and optimization, it may end up with different values in each run (local minima). Our visual inspection shows that the network finds four main groups of solutions for representing unknowns.

Fig. 9 depicts elements in $U$ for 8 selected runs that represent these four groups. In this case, $U$ has eight elements (two inputs for four neurons). Markers with similar colors represent the neutralizer values for two features (axes) and four neurons in a specific run (legends). The neutralizers in the first group (1 and 11) are roughly zero imputers in which the missing values are imputed with close to (but not absolutely) zero values. These are sub-optimal solutions that happen only in a few runs. In the other cases, the models seem to perform more than a simple imputer. In another minority group, represented by runs 52 and 100, the model turns to a mean imputer for the first feature, but it learns diverse representation for the second feature; the neurons see unknowns as borderline samples. In another minority group (represented by run 41 and 81), some neurons see unknowns as samples with extreme values (outside or close to the outer range of inputs). Run 7 and 10 represent the majority group of solutions in which each neuron has its unique perception of unknowns.

## A.6 SUPPLEMENTARY MATERIALS FOR EXPERIMENTS ON OPENML DATA

Table 2 and Table 3 summarize the OpenML datasets that are respectively used in our classification and regression analyses in Sec. 3.2. We used exactly the same datasets as in Jäger et al. (2021).

Fig. 10a and Fig. 10b compare the performance of our naive NN architecture (that are used in our experiments in Sec. 3.2) with a random forest (RF) classifier or regressor across (with default settings) benchmark datasets. The errorbars represent the variation in AUC/SMSE across 10 repetitions. The models are evaluated in a 2-fold cross-validation scheme. The random number generator seed is fixed for all algorithms in each repetition. Our results show that our simple NN architecture with two hidden layers generally performs as well as random forest models. Both NN and RF models show near the chance performance on 'pollen', 'egg-eye-state', and 'numerai28.6', thus we have removed these datasets from our analyses in Sec. 3.2. About the regression tasks, again we observed poor performance (SMSE>1) of both models in the case of 'stock-fardamento02', 'wine-quality', and 'Bike-Sharing-Demand' datasets. Thus, they are removed from our analysis.

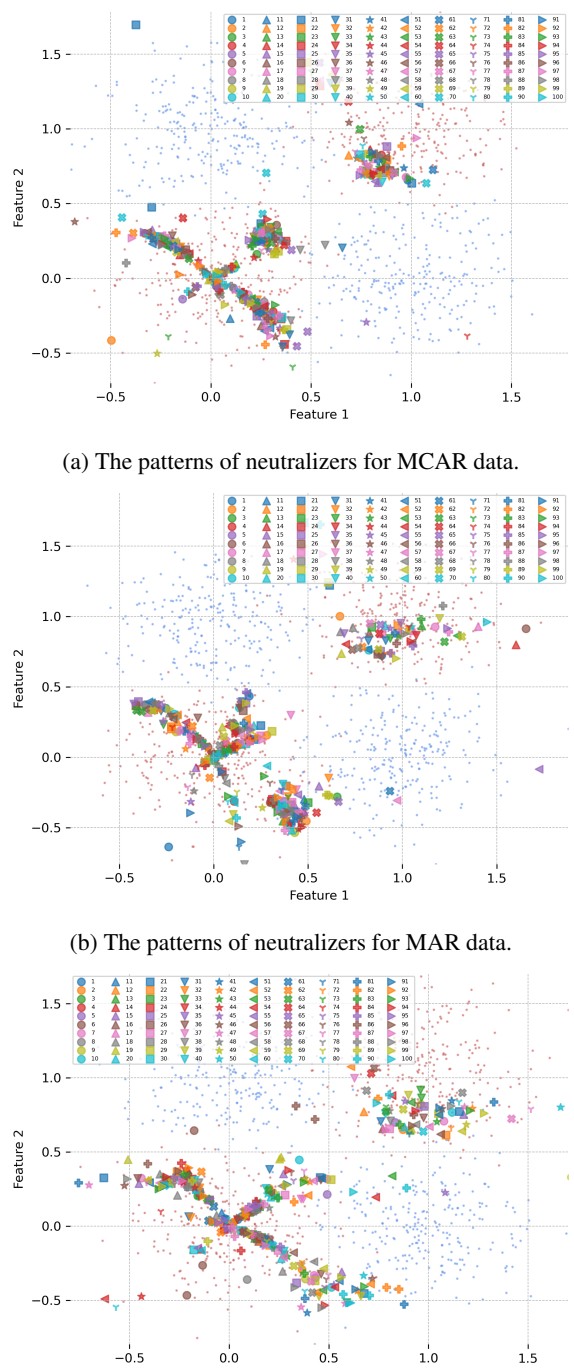

(a) The patterns of neutralizers for MCAR data.

(b) The patterns of neutralizers for MAR data.

(c) The patterns of neutralizers for MNAR data.

Figure 8: Patterns of neutralizer elements across two input features and the four neurons across 100 simulation repetitions. The markers with the same color and shape show the neutralizer values for each of four neurons in a specific run (the legends) of the pipeline. The blue and red dots in the background are representing the simulated data.

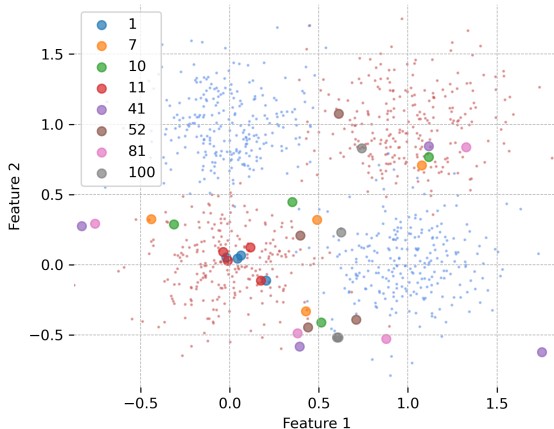

Figure 9: Pattern of neutralizers in $U$ across two input features and the four neurons. The markers with the same color show the neutralizer values for each of four neurons in a specific run (legends) of the pipeline. The blue and red dots in the background are representing the simulated data.

Fig. 11 shows the results for experiments in Sec 3.2 in the regression tasks. Plots show the difference between the SMSE of the full model (trained on complete data) and the model trained on data with missing values. As expected the difference in SMSE increases with an increase in the ratios of missing samples and features. The (m)PROMISSING shows competitive performance compared to HGB and the better performing imputers, *i.e.*, the zero, mean, and KNN imputers.

In an extra experiment, we used the MCAR scheme to simulate different ratios of samples (0.1, 0.25, 0.5, 0.9) and features (0.1, 0.25, 0.5, 0.75, 1) with missing values across datasets. Please bear in mind that applying this setting (with more than one feature with missing values) to the MAR and MNAR conditions is tricky because guaranteeing the assumptions behind these schemes can be complicated. Fig. 12 compares the performance of different data imputation methods with (m)PROMISSING on the classification tasks for different missing sample ratios (in each panel) and missing feature ratios (x-axes). The AUC drops with respect to the full model (trained on complete data) are averaged across different datasets and ten runs. The (m)PROMISSING shows competitive performance compared to the better performing imputers, *i.e.*, the mean and KNN imputers. Especially, mPROMISSING performs slightly better when the ratio of missing features is increasing. A similar overall pattern can be observed when comparing the increase in the SMSE in the regression tasks (see Fig. 13). In summary, these results show that (m)PROMISSING can compete with data imputation approaches without the need for filling the unknowns.

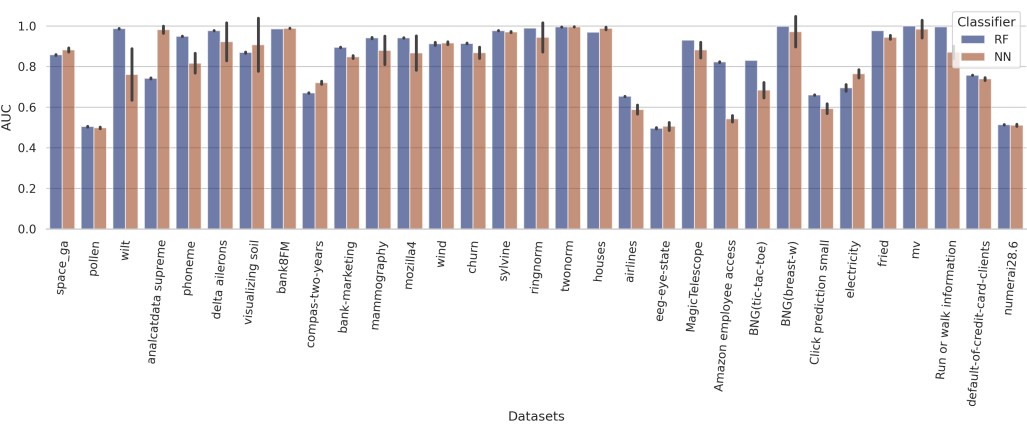

(a) AUCs in classification tasks. The higher score is better.

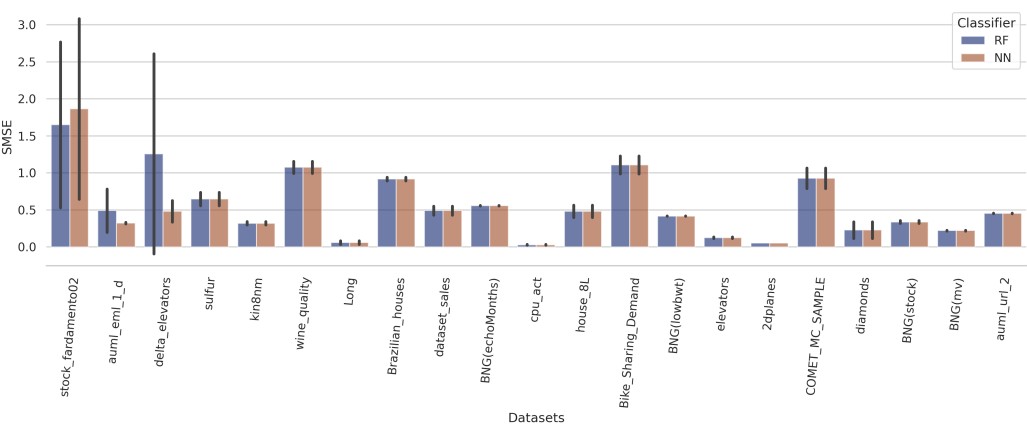

(b) SMSEs in regression tasks. The less score is better.

Figure 10: Comparison between the performance of our simple NN architecture with random forest across a) 31 classification and b) 21 regression tasks.

Table 2: Classification datasets that are used in our analysis in Sec. 3.2.

| | Dataset Name | OpenML ID | Sample Num. | Feature Num. |
|---|---|---|---|---|
| 1 | space-ga | 737 | 3107 | 6 |
| 2 | pollen | 871 | 3848 | 5 |
| 3 | wilt | 40983 | 4839 | 5 |
| 4 | analcatdata-supreme | 728 | 4052 | 7 |
| 5 | phoneme | 1489 | 5404 | 5 |
| 6 | delta-ailerons | 803 | 7129 | 5 |
| 7 | visualizing-soil | 923 | 8641 | 4 |
| 8 | bank8FM | 725 | 8192 | 8 |
| 9 | compas-two-years | 42192 | 5278 | 13 |
| 10 | bank-marketing | 1558 | 4521 | 16 |
| 11 | mammography | 310 | 11183 | 6 |
| 12 | mozilla4 | 1046 | 15545 | 5 |
| 13 | wind | 847 | 6574 | 14 |
| 14 | churn | 40701 | 5000 | 20 |
| 15 | sylvine | 41146 | 5124 | 20 |
| 16 | ringnorm | 1496 | 7400 | 20 |
| 17 | twonorm | 1507 | 7400 | 20 |
| 18 | houses | 823 | 20640 | 8 |
| 19 | airlines | 42493 | 26969 | 7 |
| 20 | eeg-eye-state | 1471 | 14980 | 14 |
| 21 | MagicTelescope | 1120 | 19020 | 10 |
| 22 | Amazon-employee-access | 4135 | 32769 | 9 |
| 23 | BNG(tic-tac-toe) | 137 | 39366 | 9 |
| 24 | BNG(breast-w) | 251 | 39366 | 9 |
| 25 | Click-prediction-small | 1220 | 39948 | 9 |
| 26 | electricity | 151 | 45312 | 8 |
| 27 | fried | 901 | 40768 | 10 |
| 28 | mv | 881 | 40768 | 10 |
| 29 | Run-or-walk-information | 40922 | 88588 | 6 |
| 30 | default-of-credit-card-clients | 42477 | 30000 | 23 |
| 31 | numerai28.6 | 23517 | 96320 | 21 |

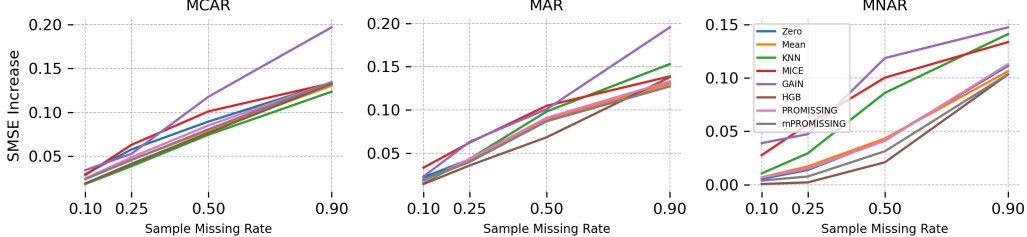

Figure 11: Comparing increase in SMSEs in regression tasks with respect to the full model across five different imputation methods, HGB, and (m)PROMISSING in MCAR, MAR, and MNAR settings.

Table 3: Regression datasets that are used in our analysis in Sec. 3.2.

|   | Dataset Name | OpenML ID | Sample Num. | Feature Num. |
|---|---|---|---|---|
| 1 | stock-fardamento02 | 42545 | 6277 | 6 |
| 2 | auml-eml-1-d | 42675 | 4585 | 10 |
| 3 | delta-elevators | 198 | 9517 | 6 |
| 4 | sulfur | 23515 | 10081 | 6 |
| 5 | kin8nm | 189 | 8192 | 8 |
| 6 | wine-quality | 287 | 6497 | 12 |
| 7 | Long | 42636 | 4477 | 19 |
| 8 | Brazilian-houses | 42688 | 10692 | 12 |
| 9 | dataset-sales | 42183 | 10738 | 14 |
| 10 | BNG(echoMonths) | 1199 | 17496 | 9 |
| 11 | cpu-act | 197 | 8192 | 21 |
| 12 | house-8L | 218 | 22784 | 8 |
| 13 | Bike-Sharing-Demand | 42712 | 17379 | 9 |
| 14 | BNG(lowbwt) | 1193 | 31104 | 9 |
| 15 | elevators | 216 | 16599 | 18 |
| 16 | 2dplanes | 215 | 40768 | 10 |
| 17 | COMET-MC-SAMPLE | 23395 | 89640 | 4 |
| 18 | diamonds | 42225 | 53940 | 9 |
| 19 | BNG(stock) | 1200 | 59049 | 9 |
| 20 | BNG(mv) | 1213 | 78732 | 10 |
| 21 | auml-url-2 | 42669 | 95911 | 12 |

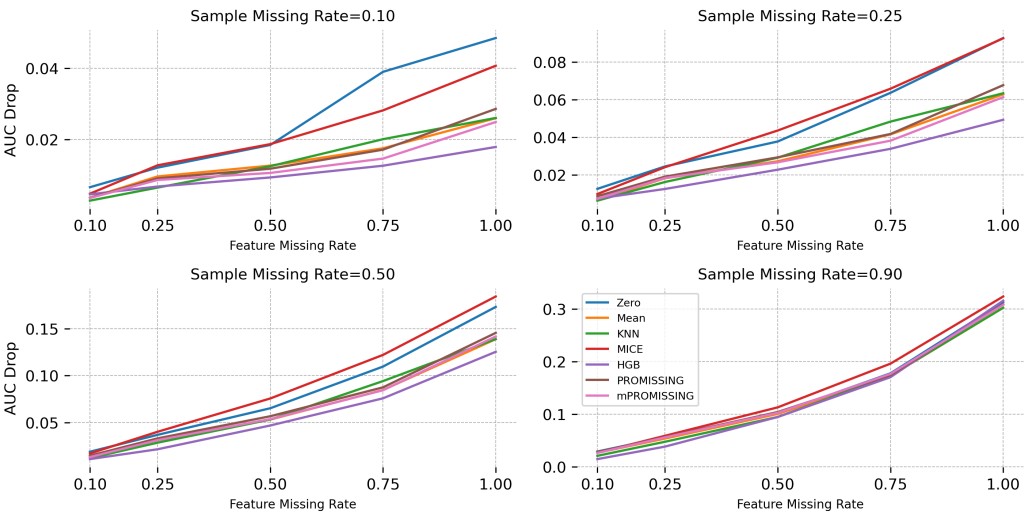

Figure 12: Comparing AUC drops with respect to the full model across four different imputation methods, HGB, and (m)PROMISSING in classification tasks.

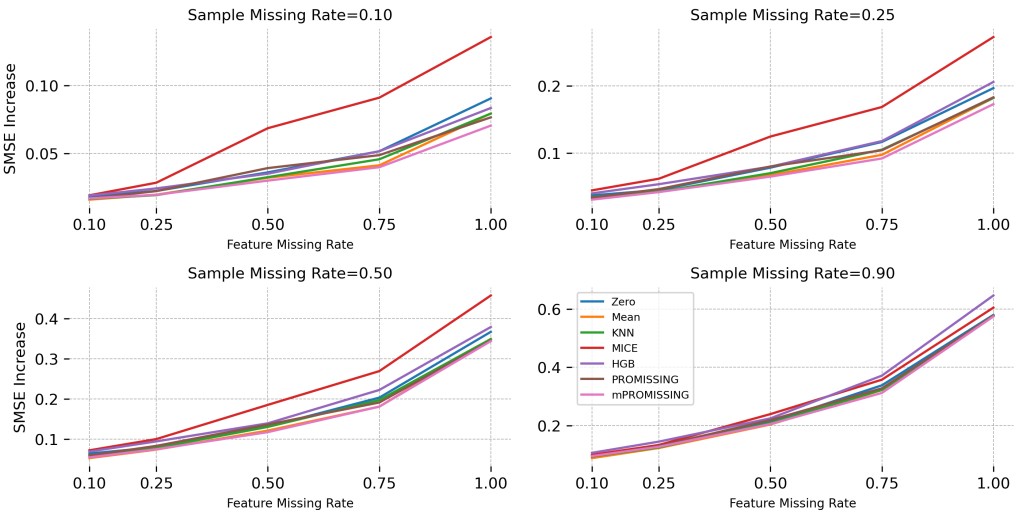

Figure 13: Comparing increase in SMSEs with respect to the full model across four different imputation methods, HGB, and (m)PROMISSING in regression tasks.

