# OpenReview forum: "PROMISSING: Pruning Missing Values in Neural Networks"
_ICLR.cc/2022/Conference — ICLR 2022 Submitted_

### Official Review · Reviewer_eatM · 2021-11-01

**Correctness:** 2
**Technical Novelty And Significance:** 1
**Empirical Novelty And Significance:** 1
**Recommendation:** 3
**Confidence:** 5

**Main Review:**

The proposed method is a very simple approach to handling missing data, and is best understood in eq 3 -- basically the bias b^{(k)} to hidden unit j has been downweighted by q/p (the fraction of observed to the total number of inputs). No justification is given as to why this transformation makes sense. (For example, one might think that one should *upweight* term \sum_{x_ in x^o} x_i w^{(k)}_i by a factor of p/q to "compensate" for the missing inputs.)

The paper does not do a good job of correctly reviewing the literature on missing data. It is unforgivable not to have a proper discussion of MCAR, MAR and MNAR in the main text (not a footnote), and to correctly point out that this classification is due to Rubin (1976) [not the reference to Azur et al, 2011].

It fails to describe the concept of multiple imputation, i.e to make use the *distribution* p(x^m|x^o) when producing predictions as p(y|x^o) = \int p(y|x^o,x^m) p(x^m|x^o) dx^m . This should nicely induce greater uncertainty as more missing data is encountered.

One other standard approach to prediction problems with missing data is to make use of the concept of the "indicator" vector m, which takes on values of 1 or 0 to indicate if an input is missing or present resp. See e.g the reference to NeuMiss networks for an example of this, although it has been in use for many years.  In fact one can propose a more general version of what the authors propose by having the concatenation (x,m) of length 2p as input to the network.  When m_i = 1, then x_i takes on a value of 0. If m_i = 0, then x_i takes on its observed value. One has weights to the k-th hidden unit from both x_i and m_i. The current proposal is obtained by having a weight of value -b^{(k)}/p from m_i. This value is shared over all inputs in the PROMISSING proposal, but in fact it would be more natural to learn the parameters of the m_i inputs unconstrained.

The paper then conducts 3 sets of experiments, on simulated XOR-type data, OpenML data, and a clinical application.

The XOR data is a simple mixture of 4 Gaussians dataset, but is perhaps a poor choice for a missing data experiment, as with only two input variables either 0 or 50% of the inputs are missing. BTW it is possible to work out analytically the best performance that can be obtained on such as dataset, using a MoG model as in Ghahramani and Jordan (1995). While both PROMISSING methods do as well as can be expected on this dataset (Fig 2b), we really should also be given results for competitor methods (zero imputation, mean imputation, KNN imputation, MICE imputation etc).

For the OpenML datasets, the given results are averaged over different datasets.  It would be most natural to consider the differences between different methods on *each* dataset, and then produce some aggregate summary. It is also essential to consider if the differences between methods are *statistically significant* -- here the 10-times repetition of experiments should help with this assessment.  See e.g.  T. Dietterich, Neural Computation 10 1895-1923 (1998) for more details on this. I note from Fig 4 that there seems to be very little difference between the PROMISSING methods and the very standard mean imputation.


Results on the clinical PPP task give AUC values on a par with the KNN results. (BTW, in answer to the text opposite Fig 5, one can use *mode* rather than *mean* imputation for categorical variables -- this is standard.)

It is also worth noting that there has been quite a lot of work about NNs for handling missing data in VAEs; this is relevant	work to	this paper.  See e.g. the HI-VAE of Nazabal et al (2020,
https://arxiv.org/abs/1807.03653), which simply "zeros out" missing inputs, "VAEs in the Presence of Missing Data" by Collier et al (2020, https://arxiv.org/pdf/2006.05301.pdf) and not-MIWAE (Ipsen et al,
2020, https://arxiv.org/abs/2006.12871). These last two papers use indicator	variable method.

Overall, this paper proposes the (m)PROMISSING methods for handling missing data in neural networks. The proposal is in fact a simplified version of the well-known indicator variable method for handling missing data, as shown above. The experiments carried out do not show any significant advantage over standard methods of data imputation.  And the synthetic experiments are carried out in the easiest MCAR case, while MNAR is of most practical interest. For the reasons above I see no case to accept this paper.

Other points:

sec 2 -- I recommend p^o and p^m instead of q and r -- it is very hard to remember what p, q and r mean.

sec 3.3.1 -- rebalancing the data to 50/50 is poor practice -- with an imbalance of only 67:33, I would not carry out any adjustment.  With a classifier trained on a different class ratio than used at test time, one should adjust the classifier post-training, as described e.g. in Bishop, Neural Networks for Pattern Recognition (1995) p 223.

Little and Rubin book -- you cite the 3rd edition, but note the first was in 1987.

**Summary Of The Paper:**

This paper tackles the problem of missing data by proposing the replacement of the input to the k-th hidden neuron in the first hidden layer from the j-th input as in eq 2. Experiments are conducted comparing this so-called "(m)PROMISSING" method with other methods for treating missing data.

**Summary Of The Review:**

Overall, this paper proposes the (m)PROMISSING methods for handling missing data in neural networks. The proposal is in fact a simplified version of the well-known indicator variable method for handling missing data, as shown above. The experiments carried out do not show any significant advantage over standard methods of data imputation.

---

> ### Author Response · Authors · 2021-11-18
> **The technical contributions in this paper are not arbitrary but based on solid linear algebra**
>
> We would like to first thank the reviewer for his/her comments and suggestions. Most comments can be accommodated by clarification and modifying/adding some text in the manuscript. The reviewer's main criticisms are around two main arguments 1) the reviewer perceives (mistakenly) the PROMISSING as a simplified version of an indicator variable, and 2) the classification/regression performance of PROMISSING is not better than simple imputers. In the following, we will discuss the reviewer's comments in detail.
> -	The reviewer thinks the neutralization process in Eq. 2 and 3 is arbitrary and without any justification. This is not true at all. We do not propose any new transformation in the PROMISSING approach. As explained in the text, our goal is to neutralize the effect of missing values on the activation of neurons. To do so, we replace the missing values with their corresponding neutralizer computed using Eq.2. In fact, Eq. 3 does not propose any transformation, but it is just a byproduct of inserting Eq.2 in Eq. 1 (see appendix A.2.1 for detailed equations). It removes the effect of weights of missing values (the second term in Eq. 1) and reduces the bias by a factor of $r/p$. If all input values are missing, then the activation in Eq.3 becomes zero (see Proposition1). If there is no missing value, Eq.1 and Eq. 3 are equivalents and it behaves like a normal NN when there are no missing values (see proposition 2 and 3). We agree that the text was not clear enough in technical details. To address reviewer comments and to avoid any further confusion about the technical solidity of our approach, we will formalize the abovementioned arguments in the form of three propositions (in the main text) with detailed proofs (in the appendix).
> -	The reviewer believes that we did not a good job in reviewing related works and strongly suggests including more discussion about missing value mechanisms including MCAR, MAR, and MNAR. Given the available space, we have tried our best to compactly cover the mainstream state-of-the-art in handling missing values in ML. Of course, we may have missed a few works in our review. We appreciate the reviewer’s suggestions and we willingly add/modify the text and references. About including more text on missing data mechanisms, we opted to only include the references rather than the full description in favor of saving some space for analysis. We agree with the reviewer that including more description might be essential for didactic purposes and we wish to have more space to do that. With current page limits, to accommodate the reviewer’s comment we can only promise to include the extra description in the supplementary. Just a quick clarification that we did not refer to Azur et al. 2011 when describing missingness mechanisms. In “The central assumption behind the MICE approach is that the missing values are missed at random (MAR) (Azur et al., 2011)” Azur et al. is referred to as a reference for the whole sentence and not for MAR. We will modify the text to avoid further confusion. We will also include Rubin (1976) and suggested studies on the applications of deep generative modeling on missing data in our bibliography as suggested.
> -	We agree with the reviewer that multiple-imputation offers a mechanism to induce uncertainty in the predictions when facing many missing values. Our criticism of multiple-imputation is their high computational complexity and their MAR assumption (as mentioned in the text). We have included several references for multiple imputation techniques. We believe the manuscript plus its references is self-content. We hope the reviewer understands that it is very difficult to fit the description of every mentioned method in the limited available space.
> -	The reviewer believes using classic XOR data with two variables in our simulation study is a poor choice due to its simplicity. We disagree with the reviewer on this point because we favor the simplicity of this problem as it provides the possibility to better analyze the behaviors of the proposed method. Benefiting the simplicity of the problem we can easily achieve the Bayes classifier performance using a simple model (one layer with 4 neurons). On the other hand, the fact that the problem is two-dimensional makes the visualization of the 2D patterns of neutralizers feasible. To address the reviewer's comment, we will include the performance for zero, mean, and MICE imputations (the KNN is already there) in Fig. 2. We furthermore included the results for MNAR and MAR in this section. The results for MNAR data are indeed very promising and show that PROMISSING provides the possibility to jointly model data and missing value mechanism (without adding extra complexity to the model by extra parameters and hyperparameters) that is an extremely favorable feature.

---

> ### Author Response · Authors · 2021-11-18
> **The PROMISSING differs from indicator vector, the core contribution in this paper is not improving the performance**
>
> - The reviewer believes using an indicator vector when modeling data with missing values is a more general case for PROMISSING. We believe the reviewer did not grasp some aspects of the presented methods (why and how neutralizers are used). We strongly disagree that PROMISSING is a simplified indicator because:
> 1) Using an indicator vector is not a method for handling missing data per se, but it is a complementary approach that can be used in combination with different data imputation approaches for better performance, for example, in MNAR scenarios. In other words, to be able to use indicators we first need to use an imputer for filling the missing values. The reviewer, in his/her example, suggests using a zero imputer (that might not be efficient for binary or categorical features). We can combine the indicator approach with different data imputations at the modeling stage of course by compensating for higher model and computational complexity.
> 2) We believe the underlying mechanics for handling missing values in PROMISSING is quite different from the data imputation + indicator vector approach. The PROMISSING differs from this as there is no need to use any kind of data imputation, the NaNs remain NaN. Therefore, the performance of models based on indicator vectors highly depends on the data imputation quality. Furthermore, unlike data imputation with an indicator that doubles the number of parameters of the model (we will have $2 \times p$ parameters), the number of parameters of the model remains unchanged when using PROMISSING. Thus, PROMISSING is computationally more efficient and at the same time comes with less parameter uncertainty on small sample size data (this is a crucial feature when we have access to few high-dimensional samples).
>
> - We completely agree with the reviewer that plotting the average performance drop across datasets is not the best representative way to present the results on OpenML data. We have already tried several ways (e.g., seaborn catplots) but given the number of benchmarked approaches and different missing value rations (for both samples and feature), it is very difficult to represent these results compactly. The reviewer proposal does not work because the difference between the performance of the two methods could be negative or positive (thus bimodal) that making summarization cumbersome. Putting the suboptimality of the current plots aside, we still think the graphs are representative enough of our conclusion that “the performance of PROMISSING method is competitive with widely used imputers”. We never claimed in the paper that it does a better job. In fact, we do not believe in a universal method for handling missing data that works the best on all problems. We believe the right choice for the imputer strongly depends on the specifications of data and the missing value mechanism.  The PROMISSING works better in almost a third of the benchmarked datasets, but we intentionally avoid only presenting better performances. Instead, we would like to direct the attention of the reviewer to our core contributions in this manuscript, where on several experiments on more than ~50 datasets, we empirically show that our method is 1) can approach the performance of the optimal Bayes classifier without the need for data imputation, 2) learns meaningful representation for unknowns, 3) performs as well as baseline imputation methods, 4) is more reliable in uncertain environments, 5) can be used for counterfactual interpretation of complex ML models.
> - The reviewer criticizes the similar performances of PROMISSING and KNN in the PPP task. Continuing from the last comment, we again emphasize that our goal is not to achieve the best performance. As thoroughly discussed in the text (see results in Fig 6 and 7), the main advantages of using PROMISSING in comparison to data imputation lie in its 1) more reliability in decision making when facing many unknowns, and 2) more flexibility for model decision explanation using counterfactual interpretation. And believe us, these two features make the actual difference in clinical applications (not 0.05 better or worse classification performance). Furthermore, our method is distinguished from its alternatives as it avoids imputing data. This is a crucial feature in delicate applications of ML in clinical, robotic, and autonomous machine control. Imputing unknowns can be extremely dangerous with catastrophic consequences (just imagine imputing the data of a failed sensor in airplane autopilot). The reviewer also proposes to use mode for imputing categorical variables. We cannot pretend that we are not disappointed by the sarcasm behind this comment, but the text next to Fig. 5 says “Note that, due to binary and categorical features, many imputation techniques (such as mean and iterative imputer) cannot be directly applied to our data.”. This does not mean that there is no imputer to handle categorical missing values.

---

> ### Author Response · Authors · 2021-11-18
> **Final words**
>
> In the end, we found the evaluation of the reviewer on the technical and empirical novelty of our work quite unfair, and we think many aspects of our technical and empirical contributions are overlooked. Please bear in mind, improving the performance of the model is not the core contribution in this work and according to the ICLR review policy, the quality of submissions should not be judged merely on improving performance but novelty, relevance, impact, and added knowledge should also be considered (https://iclr.cc/Conferences/2022/ReviewerGuide). We hope our responses and clarifications besides the new empirical results provide a basis for a more positive evaluation of our contribution. We strongly believe in the significance of our technical (the first principled approach for handling missing data in NNs without imputation and without extra parameter and hyperparameter) and empirical (extensive evaluation of different aspects of the proposed method) contributions and considering its simplicity we are sure about its future impact regardless of the outcome of this review process. We will submit soon the revised version, meanwhile, we of course very welcome further reviewer’s comments and suggestions for improving our work.

---

### Official Review · Reviewer_Bfyk · 2021-11-01

**Correctness:** 3
**Technical Novelty And Significance:** 4
**Empirical Novelty And Significance:** 2
**Recommendation:** 6
**Confidence:** 5

**Main Review:**

The manuscript presents a novel approach to dealing with missing values, the architecture they propose is a modification on the fully connected layer which allows the user to plug this layer into any neural network. This makes this method particularly versatile since it can have many uses beyond the ones presented in the manuscript. The methodology of the paper is well written and the methodology is mathematically sound. Experiments and results sections could use an improvement to enhance clarity given the heterogeneity of analyses conducted. That being said, there are some comments that I think should be addressed:
Major comments:
- The baselines you compared your method to contain only a few state-of-the-art methods. I would add the iterative imputer with ridge regression, and add a few of the newer methods that are well performing. I would suggest GAIN (Yoon 2018, last in the references) and GRAPE (You et al. 2020 arXiv:2010.16418) as state-of-the-art baselines.  In addition, you could also compare to an input where the missing value flags are inserted as an input to the NN which is an approach that is widely used that also forgoes imputation. This would give the paper more strength especially to make sure that your model outperforms others in the conditions mentioned.
- The results in figure 2 show overlapping error bars. First, it is important to specify the error bar meaning (in other figures as well) and also cannot claim that the PROMISSING method surpassed the kNN method given the overlapping error bars unless you have performed an appropriate statistical test.
- It is unclear to me why different imputation methods are being utilized in different experiments (kNN is used only for the psychosis dataset while 4 methods are utilized for simulated data) and also different result visualization methods (AUC vs AUC drop vs model trajectory).  I understand that you are trying to make a different point each time but it is natural to discuss the general performance on the psychosis task before the trajectory. One possibility is that the imputation is able to generate better predictions even in the absence of a large number of modalities.
- Additionally, it appears from figure 6 that there is a higher number of patients with positive output than with negative output which could be the reason why the regular network is showing the positive bias when the number of missing modalities is high. While there are mitigation strategies for that in regular neural networks, this could also be a strength point for your model. It is important to report the ratio of imbalance as it would explain why the baseline model ends up with a bias as you remove more features and then you can discuss how your model mitigates this.

Minor comments:
- There is an assertion the methods that PROMISSING the features are not inferred from observed features causing the MAR assumption to be relaxed. This is not entirely accurate given that the training phase will be affected by the missing values so it will still work best when the data is MAR similar to how regression-based imputation methods perform best at those conditions. Also, having the data MNAR makes the problem of inference easier but less generalizable to patterns of MNAR data of patterns never seen before. I would suggest removing this part and replacing by a statement about how this method forgoes imputation steps which can be costly in the case of complex imputation mechanisms.
- In section 3.1, it is mentioned that the RandomForest Regressor is equivalent to MICE, does that mean you used multiple imputation (data augmentation) or used a single imputation variant? Also citation of the paper for this method is missing (https://doi.org/10.1093/bioinformatics/btr597)
- For the OpenML dataset, it is not mentioned what data imputation method is utilized with the RF benchmark. In addition, in table 2, there are some datasets that have zero features, is that a typo?
- There are many missing technical details in the testing procedures that I would suggest to be added to the supplementary info such as the training/test split ratios, hyperparameter-selection procedure (if exists), number of training epochs... etc. This information is crucial for reproducibility.
- Regarding figure 3, the figure even after simplification is quite difficult to comprehend because of the missing distinction between the 8 different weights. I suggest choosing one neuron and plotting the 2 weights that feed into that (or even one) this would decrease a lot of clutter. Also maybe it is good to plot the original weights without missingness to visualize the effect of missingness.
- I am unsure of how (un)balanced the datasets used here are. It is beneficial to report those values because for highly unbalanced datasets (quite common in medical data), other measures such as precision and recall are much more valuable and representative of the performance.
- In figure 4, the MCAR scheme used to artificially remove values is unclear to me how it was achieved. I am guessing that feature missing rate meant the maximum number of features that had missing values and sample missing rate is the amount of missing values that are missing in a given sample. I would first like to confirm if that were true. Additionally, there are some points that need clarification here: 1/ Is that sample missing rate calculated as a percentage of the feature missing rate? 2/ Are the features removed constant across samples or do you randomize that choice over samples as well?
- Second question above can also be posed at the modalities removed in figure 6.


**Summary Of The Paper:**

The paper provides a novel method of handling missing values in neural network architectures by replacing the missing input with a pruning signal that causes the network to utilize the existing values only. The author(s) tested the method on simulated data as well as real data on a group of classification and regression tasks.

**Summary Of The Review:**

The method presented in this paper is novel and presents an interesting handling of the missing data problem. It can even be tested on DNN-based imputation such as generative models. Given the versatility of the model, it has so much potential for many uses. The main weakness of the model is the lack of significant improvement in performance in comparison to the baseline models but this should also be commended as the author(s) did not cherry pick the results where the model is well-performing and rather opted to report all their results. The other weakness is the small number of baseline models the new method is compared to. I believe the authors should summarize where their method shines vs where it does not in the discussion or summary because the way the method is pitched in the current format is that it leads to better performance than the imputation-based methods in the basic tasks of classification/regression while the results do not reflect that. Additionally more baseline methods should be added to the comparison to enable proper comparison.

---

> ### Author Response · Authors · 2021-11-18
> **Response to Major comments**
>
> We would like to first thank the reviewer for the positive comments on the novelty, clarity, and mathematical soundness of our approach. We feel very satisfied reading the reviewer’s comments, as it seems the manuscript is well-read and understood. The reviewer’s comments include a request for extra experiments (comparison with more baselines), six clarification questions, and three suggestions. The majority of comments are applicable and will be applied to the revised version. In the following, we will discuss the raised points in detail.
>
> **Major Comments**
>
> - The reviewer requests for comparison with more baseline approaches including GAIN, GRAPE, MICE with ridge regression, and when using indicator vectors in combination with different approaches. To address the reviewer's comment, we have included the comparison with the GAIN method in the revised version. We needed to resolve quite a few technical difficulties because the implementation for GAIN needs outdated Tensorflow 1 while our approach works with Tensorflow 2. The GAIN method, despite its complexity, is among the worst performers, especially in the MNAR setting. This is not a surprise and is reported in other ICLR submissions recently (see for example https://openreview.net/forum?id=_MRiKN8-sw ). We could not, unfortunately, gather all the results for the GRAPE model yet and we cannot promise to have them on time (at least at this stage) but we do not expect a major surprise. About the ridge regression, we already use Bayesian ridge regression in our experiments with the iterative imputer (referred to as MICE in the new revision). We agree with the reviewer that combining the indicator vector with imputation might improve the results, but please bear in mind this is also the case for PROMISSING (as the cost of doubling the parameters of the network). In this paper, we aim to compare the pure imputation approaches in their default settings (thus we intend to remove any external factors that might affect their performance including parameter tuning and additional inputs such as indicator vectors). In the end, we would like to emphasize that we never intend in our paper to sell our method as the best method in terms of performance. This is not the goal. When comparing with different methods, we only show that our method provides competitive results with respect to imputation techniques without the need for data imputation. We think this is already an advantage to skip imputation while getting the more or less similar performance.
> - The error bars in Fig. 2 represent the standard deviation around the mean performance across repetitions. We have clarified this in the text in section 3.1. As suggested by the reviewer, in the revised version we will use a statistical significance test (Wilcoxon rank-sum test) when comparing methods.
> - The reviewer asks why only KNN is used in the PPP experiment. As it is already explained in the text (3.3.1), the OPTiMiSe dataset contains many categorical and binary features. This makes it impossible to use a range of imputation techniques such as mean, MICE, and GAIN by default. We have briefly mentioned this in section 3.3.3. This can be seen as another advantage of our method that simplifies all these complications as it does not need data imputation. About the consistency of the evaluation metric, we always use AUC for classification tasks and SMSE for regression tasks. In experiments on OpenML data, we are interested in the performance drop (not the performance itself) across different methods, datasets, and missing ratios. Therefore, the AUC drop or increase in SMSE is plotted. In our opinion, using the performance drop even makes more sense when we are aggregating results over many datasets (because of the different range of metrics across different datasets). About the PPP experiments, we have presented the classification results (AUCs are in section 3.3.3) before discussing the trajectories of model predictions.
> - Regarding class unbalance, we agree with the reviewer that class unbalances might bias the predictions toward the majority class. But we have already addressed this by balancing the training set. As it is mentioned in the text (in section 3.3.1, “Considering the unbalanced class distributions (67% of patients get remitted), in each fold of K-fold cross-validation, we resampled the minority class samples in the training data to match its size to the majority class.”), we have balanced the samples by resampling the minority class. Just for clarification, the results in Fig. 6 (it is Fig. 4 in the revised version) are for the 17 test subjects (not training subjects), so their class distribution does not affect the classifier bias toward a certain class.

---

> > ### Comment · Reviewer_Bfyk · 2021-11-22
> > **Clarification of performance comments**
> >
> > Thanks a lot for your responses and the revision. It is much improved in my opinion.
> >
> > There is however a sticking point in your response regarding the claim that the method is not claimed to improve on the existing methods but rather provide a simpler architecture. While the statement is true, it is always important to show that the method is still good enough compared to imputation methods which is why I requested the other imputation methods to make sure that there is not a method that actually outperforms your method strongly. I was hoping to have more methods implemented than the GAIN. This is also why I see the drop is AUROC and AUPRC as not a very good metric since the baseline performance is still important and it will not make the plot any more complicated. I understand that the page limit makes it difficult to accommodate the responses. I just wanted to point these considerations out for the authors since the responses focused too much on that performance point. Some reviewers would also argue that the contribution is not significant enough to be accepted which is something I don't fully agree with myself but again pointing out what I understand from other reviewers.

---

> > > ### Author Response · Authors · 2021-11-22
> > > **Thank you for your positive evaluation of the revised version**
> > >
> > > Thank you for the reply and your positive evaluation of the revised draft. We also believe that the revised version is more robust.
> > > - Regarding the performance, we completely agree with the reviewer that "it is always important to show that the method is still good enough compared to imputation methods". This is actually why we have conducted our OpenML experiment on ~50 datasets. To show that our method works on par with its alternatives WITHOUT the need for imputation (this is already a huge contribution) and WITHOUT adding extra complexity to the model. We think our results are in accordance with this statement: our model is as good as its imputation alternatives.
> > > - Regarding other methods, the reviewer suggested GAIN and GRAPE. The results for GAIN are already included in the text. The experiments on GRAPE are running now and the upcoming results are not better than even a simple mean imputation (in GRAPE we needed to also use an indicator vector that is expected to help a bit with performance but it did not). It is also surprising to us that even simpler methods such as KNN or mean imputers work on par with these complex imputers. We can include the GRAPE results in later stages in case of a positive outcome of the review process.
> > > - As explained in our response, the AUC drop is computed by subtracting the BASELINE performance (the performance of the model on complete data, so no imputation is needed and there is only one baseline model for all methods) by the performance of the model trained on the imputed data. We found this more intuitive because as the reviewer suggests the performance of the baseline model is considered while commuting the drop. The reviewer suggests summarizing only the AUC of models trained on incomplete data. We can include this also in the appendix in later stages but it has no effect on the conclusions of this paper (Please just bear in mind that in this case the baseline performance is not considered).
> > > - We strongly believe in the significance and impact of this work regardless of the outcome of the review process. We plan to implement promising for other NN layers (CNN and LSTM, etc.) and include implementation in major DL packages (e.g. Keras). PROMISSING is not a new architecture or model, it is a general approach for NNs based on the NN theory. As explained in the text, it can be implemented by adding 4 lines to the implementation of NN layers and it can be used only by changing a flag (e.g., naninputs=True). It can be used with any architecture without changing anything and just by setting a flag. This is while we believe it will be used a lot in the future. This is to us a real impact and significance.
> > >
> > > We hope our response resolves all the reviewer's concerns. We are open to receiving more questions and feedback.

---

> ### Author Response · Authors · 2021-11-18
> **Response to minor comments**
>
> **Minor Comments**
> - We agree with the reviewer that the statement about the relaxation of the MAR assumption is misleading. Here we intended to convey the message that: Unlike some regression and model-based approaches that assume (apriori) MAR on the missing data mechanism (thus fail if the missing data are not MAR), handling missing values in PROMISSING (neutralizing their effect on the activation of the neuron) does not assume any specific missingness mechanism. To address the reviewer’s comment, we will modify the sentence to avoid further confusion. Furthermore, in the new version, we will provide empirical evidence for the effectiveness of PROMISSING in the MNAR setting.
> - We never say that random forest regression is equivalent to MICE. But random forest can be used as the regressor within the MICE imputer. To avoid further confusion, and for more consistency, we removed the sentence and always used Bayesian ridge regression (the default) for the MICE imputation. For implementing MICE, we have used the multiple imputation procedure implemented within iterative imputers in the scikit-learn package.
> - In OpenML experiments, the random forest model is only used on full data (so no imputation is required) to show the reasonable performance of the simple NN architectures used in the experiments (bar plots in the appendix). The random forest was never used in the analysis in the main text. Thank you for pointing out the typos in table 2. Indeed, they are not zero and will be corrected in the new version.
> - The reviewer asks for adding technical details (training/test split ratios, hyperparameter-selection procedure, number of training epochs) of the experiments to the supplementary materials. There should be a misunderstanding as we think all the requested information is already included in the main text. For the simulation study, the half-split is used for training and test sets (section 3.1). We used 2-fold and 10-fold cross-validations for the OpenML and PPP experiments, respectively (see section 3.2 and 3.3.3). As mentioned in the text, all models and imputations are applied with their default settings, therefore, no hyperparameter selection is performed. About the number of epochs, as it is mentioned in the text, it is 1000 for simulated study (see the x-axis in Fig.2) and 100 for OpenML (see text in section 3.2).
> - We agree that understanding the patterns of neutralizers in Figure 3 is a bit tricky. We have tried a few ideas to make it more intuitive but all have failed; thus, we have modified the text and moved the figure to the supplementary. The message here is that the PROMISSING model can learn different imputers from data (some neurons become zero imputers and some others become mean imputers, and some perform more than a simple imputer).
> - Yes, the OPTiMiSe dataset is an unbalanced dataset with 67 percent of patients getting remitted. This is already mentioned in the main text in Section 3.3.1 (“Considering the unbalanced class distributions (67% of patients get remitted),”).
> - The missing feature rate refers to the proportion of features with missing values, while the sample missing rate refers to the proportion of samples with missing values. For example, if we have 100 samples with 10 features, then a 10% feature missing rate and 50% sample missing rate means we have missing values for 50 samples and each of these samples misses 1 feature (and this feature is fixed across samples). Due to the change in experimental settings (now we also include results for MAR and MNAR), we have moved these results to supplementary. Now we assume always one feature is missing (the most informative one).
> - The removed modalities in figure 6 are fixed across samples in each repetition.
>
> In the end, we would like to reemphasize that we do not intend to sell our method as the best-performing method in terms of accuracy. We do not believe in a universal solution that always works best for all problems especially for the very difficult problem of dealing with missing values. Based on our experience, we strongly believe the optimal solution is always problem and data-dependent. Therefore, we have tried to be very careful about our claims in the text (we always say PROMISSING provides competitive performance compared to others). We appreciate it if the reviewer points us to a sentence or statement that suggests otherwise, and we will willingly modify it. We hope our additions and clarifications provide a basis for a more positive evaluation of our work. We will submit the revised version soon and meanwhile, we appreciate the reviewer’s further suggestions.

---

### Official Review · Reviewer_FmEA · 2021-11-01

**Correctness:** 3
**Technical Novelty And Significance:** 4
**Empirical Novelty And Significance:** 3
**Recommendation:** 6
**Confidence:** 4

**Main Review:**

Strong points:
- The paper is well-motivated and the method is interesting.
- I appreciate that the authors have included a NN architecture implementing the proposed approach for use with a popular NN modeling software.
- The data analyses are fairly thorough, but could be improved (see below).

Major concerns:
- There is a large literature on weighting methods (e.g., inverse probability weighting, doubly-robust estimators) that was ignored in this paper. You should include some references to this literature and discuss how your proposed method is related (or not) to these methods.
- What is the motivation behind the 'neutralizer' value? How should we interpret this?
- I'm not convinced that keeping unknowns as unknowns allows the MAR assumption to be relaxed (page 3). If there truly is data missing-not-at-random, then it seems *any* approach suffers from this, regardless of whether or not the missing mechanism is modeled (which is a difficult or impossible task with MNAR data).
- The take-home message from the simulations and data analyses, in my view, is that PROMISSING does no worse than methods that use imputation. This is at odds with the motivation, which describes potential for increased performance over these other methods. Why do we not see improvements in the experiments and analyses?
- In the simulations:
    - what does performance look like if the data are MAR? or MNAR?
    - how does PROMISSING perform with no missing data (i.e., compare PROMISSING/mPROMISSING trained without missing values to the NN trained without missing values -- are they the same?)?
    - why is there such a large decrease in predictive performance when the test set has missing values vs when it doesn't (Figure 1)?
    - 50 Monte-Carlo replications is quite small (section 3.1); consider increasing to at least 500
    - I find it difficult to digest Figure 3, and I found the discussion on page 6 unsatisfying. Consider re-framing this paragraph or removing the figure.
- In the data analyses:
    - again, why did you focus on MCAR? did you consider MAR or MNAR?
    - again, 10 replications is too small, run at least 100
    - Figure 4 is difficult to digest, because the lines are too close together. What are the magnitudes of the difference in performance between approaches? What is the Monte-Carlo standard error? I suspect that the differences between methods are quite small relative to the error.

Minor concerns:
- You could make more clear that equation (1) is the traditional approach to NN activation, while your approach augments this. At first, it is confusing to say that you need imputation to fit (1) when in reality you fit a different objective.
- Check for typos and language use throughout.


**Summary Of The Paper:**

In this paper, the authors propose a method titled PROMISSING; this provides a new approach to handling missing data. Rather than imputation, a complete-case analysis, or inverse probability weighting, among other methods, the authors advocate for learning a problem-specific numerical representation for unknowns. The approach is interesting, and the experiments and data analyses are a good start at understanding the method.

**Summary Of The Review:**

I vote for accepting the paper subject to some additional experiments, based on my review above.

---

> ### Author Response · Authors · 2021-11-18
> **It is not all about performance, we added further experiments on MAR and MNAR settings**
>
> We would like to thank the reviewer for the positive comments on the motivation and implementation of our work. The reviewer comments include 6 clarification questions and 2 additional experiment requests. All comments and suggestions are accommodated in the revised version. Including:
> -	The reviewer suggests including references and discussion on applications of inverse probability weighted estimation approach in handling missing values. We thank the reviewer for introducing us to this interesting approach. The most interesting aspect of the IPW is that it does not rely on data imputation (similar to our approach). We have included two major references plus a very brief description of this approach in our introduction (two sentences). We are sure, our short description barely touches the topic, but this is the maximum we can do to accommodate both reviewer’s suggestion and the page limit imposed by the conference.
> -	The reviewer asks about the motivation behind using the ‘neutralizer’ and its interpretation. As it is mentioned in the second paragraph of the method section, the motivation for using a neutralizer is two-fold 1) it prunes the missing values from inputs of a neuron, 2) it neutralizes the effect of missing values on the neuron’s activation (by canceling the second term in Eq. 1). The pruning ensures that the neuron receives a numerical value and can function properly. However, these values are not at random or based on imputation. Instead, they are derived from parameters of the network (weights and bias, see Eq. 2) thus they are learned during the optimization process from data. As you can see by inserting the values for the neutralizers (Eq. 2) in the activation of a neuron (Eq. 1) the effect of missing values is eliminated on the activation of the neuron (Eq. 3). Thus, the neuron will try to use only the non-missed values to compute its activation. To interpret, the patterns of neutralizers we visualized them in Fig. 8 and 9. As it is discussed, PROMISSING provides the possibility for the neurons to learn neuron-specific imputers from data. In our simulation study, some neurons learn to be zero imputers and some learn to be a mean imputers (note that this can be different from a neuron to another thus it differs from constant imputation).
> -	About the relaxation of the MAR assumption, we agree that the previous draft was failing in presenting some empirical evidence on the effectiveness of the PROMISSING approach. As suggested by the reviewer (in the next comments), we have included other missing value mechanisms in our simulation and OpenML experiments. Our results show that PROMISSING provides the possibility to learn the patterns of missing values in data thus may be more effective in the MNAR setting.
> -	About the take-home message, we never intended in the manuscript to convey the message that PROMISSING provides a ‘better’ classification/regression performance compared to imputation approaches. If the reviewer directs us to any sentence suggesting this message, we will willingly remove it from the manuscript. Instead, and throughout the manuscript (abstract, introduction, conclusions), we claim that PROMISSING provides similar or competitive performance compared to other methods without the need for “data imputation”. In fact, keeping unknowns as unknowns enables us to provide more reliable predictions in delicate applications of ML in, for example, clinical decision making (such as in the PPP case as demonstrated in Fig.6). We believe, at least in the MCAR setting in which there is not a certain pattern in missing values, it is even very suspicious if one reports significantly higher performance when giving less information to the model (as is the case for PROMISSING).
> -	Following the reviewer's suggestion, we have included the MAR and MNAR settings in our experiments on simulated data and OpenML datasets. Our results on simulated data show the promises of the PROMISSING approach in the MNAR setting. However, our conclusion in the OpenML experiment remains the same: “PROMISSING provides competitive performance to its alternatives without the need for data imputation”. Please bear in mind that in the OpenML case we intentionally aggregate the results across many benchmark datasets to ensure that our conclusion is not dataset-specific. Of course, PROMISSING performs better than other methods in a subset of benchmarked datasets, but we intentionally avoid cherry-picking (nobody would have complained if we had just reported the results of 10 instead of the full 28 datasets with PROMISSING performing the best).

---

> ### Author Response · Authors · 2021-11-18
> **A PROMISSING neuron applied to data without missing values performs just like a normal neuron**
>
> -	The reviewer is interested to know how the PROMISSING performs when there are no missing values in the inputs. The answer is simple “It performs like a normal NN.” This is another appealing feature of our approach that it does not change anything unless needed. For example, the functionality of our ‘nanDense’ layer is very similar to the ‘nanmean’ function in numpy or MATLAB. Their outputs remain the same as a ‘Dense’ layer or ‘mean’ function if there are no ‘nan’ in inputs. In the PROMISSING case, the activation of neurons is computed using Eq.3. If there is no missing value in inputs ($x^m= \varnothing$), we have $\frac{q}{p}=1$ thus the activation of neurons in equations Eq. 1 and Eq. 3 are exactly the same. This is also the case for the activation of mPROMISSING in Eq. 5 (there was a typo in Eq. 5 in the submitted version that is resolved), then the input for the compensatory weight is $\frac{r}{p}=0$ thus the effect of $w_c$ is eliminated and Eq. 5 and Eq. 1 are equal. To address the reviewer's comment, we have emphasized more on these features of the PROMISSING approach by including three propositions in the main text and their proofs in the appendix. For more clarity, we have also included the detailed derivation of Eq. 3 from Eq. 1 and Eq. 2 in the appendix.
> -	The reviewer is curious about the performance drop in Figure 1 when the test set contains the missing values. This is an interesting observation. We call this the ‘test bias’ and will discuss it in the revised version. This can be seen as a measure of the negative effect of missing value handling strategies on the generalization performance of the model. In fact, the test bias measures how the generalization performance of models is affected by the data degradation imposed by each strategy (see Fig. 6 on how the quality of original XOR data is affected by imputation). Our experiments show that in the MCAR and MAR scenarios, m(PROMISSING) methods show less test bias compared to other methods.
> -	The reviewer asks to increase the Monte-Carlo repetitions from 50 to 500 in the simulation experiment (also from 10 to 100 in the OpenML experiment). We agree with the reviewer that increasing the number of repetitions may result in a more accurate estimation of performances. To address the reviewer's comment, we have increased the number of repetitions to 100 in the simulation study, but since the difference (from 50 to 100) was not tangible we have stopped it at 100. We will include the new results in the revised version. About the OpenML experiment (it is a huge experiment), we needed to also perform the experiments for MNAR and MAR settings. Thus, in favor of time and in order to have the results ready before the deadline we could not repeat the experiments more than 20 times. Again, the difference (with 10 runs) is negligible.
> - Figure 3 (in the first submission) shows the patterns of neutralizers across several runs. It shows that the network can learn different types of imputers from data ranging from a zero imputer (a local minimum that happens only in a few runs) to a mean imputer. This analysis directly relates to the second comment of the reviewer on “what is the interpretation of neutralizers?”. To accommodate the reviewer's suggestion, we have simplified the text to the core message and moved the figure and detailed description to the appendix.
> - Yes, in Figure 4 as is mentioned in the text almost all methods perform similarly. And therefore, we do not conclude that PROMISSING performs better than the other methods, instead, we claim it provides a very competitive performance. The measures in this figure are summarized across datasets and replications. So, it is very difficult to compare them using the Monte-Carlo error (it will only tell us which method performs better in one dataset while we have 46 datasets). Based on our observation and a couple of recent studies, the best imputation (or modeling approach) for handling missing data highly depends on two factors 1) missing data mechanism, 2) the data itself. Due to the second factor, it is very difficult to suggest the best method. Some methods work better on some datasets. PROMISSING works better in a subset of these datasets but we opt to report the full results. To accommodate reviewer comments, we will revise any sentence that suggests PROMISSING is better.
>
> We appreciate again the reviewer’s constructive comments, and we think the text is clearer and the experimental setups are more solid after the revision. We hope our clarifications and experimental additions can convince the reviewer for a more positive evaluation of our work. Supported with our experimental results, we strongly believe this work is a good addition to the community as it provides a simple yet principled method to deal with missing values in neural networks. We will upload soon the revised version, meanwhile, we appreciate the reviewer’s further comments.

---

### Official Review · Reviewer_DWUQ · 2021-11-02

**Correctness:** 3
**Technical Novelty And Significance:** 4
**Empirical Novelty And Significance:** 3
**Recommendation:** 6
**Confidence:** 4

**Main Review:**

## Strong points
- The computationally and conceptually light approach to handle missing values in neural networks appears to be an interesting alternative to overly simple imputation (such as mean or 0 imputation) and more costly imputation strategies (iterative imputation, multiple imputation). The empirical findings suggest that it performs at least as good as more complex imputation strategies, at least in the MCAR case, which is encouraging.
- An important and application-oriented aspect of this approach is that it seems to reflect the uncertainty of the missing values in the predictions (e.g., the probabilities in the case of binary classification in their clinical example get closer to 0.5 as the amount of missingness increases). This is an interesting feature and I would encourage an analytical/theoretical assessment of this aspect in a revised version or future work in this direction.
- The experiments and results of Section 3 are well presented and commented and the details provided on the method and the simulations allow for easy reproducibility of the results.
- The article is well written and provides an adequately succinct yet sufficient bibliography of missing values handling in statistics and machine learning.

## Issues/Points that require clarification
- From the introduction and the description of the experiments it does not become clear to me which case this method is most suited for. To my understanding, PROMISSING should be applicable in cases where classical (conditional) imputation would also work, i.e., when the data is MAR (or maybe also MNAR?) and the outcome does not depend on the missingness pattern itself. In this case the outcome $y$ is defined via the complete data and not the incomplete data. This could maybe be added explicitly to distinguish this paper's setting from the case of potentially predictive missingness patterns (Le Morvan et al., 2021).
- The patterns of neutralizers presented in Figure 3 suggest that certain missing values are detected as "predictive" of the outcome (runs 7, 9, 3 and 5). It would be interesting to rerun the experiments and plot a similar figure but in the case of MNAR data where the missingness pattern can indeed be predictive (to some extent) of the outcome. This point relates to the previous comment about the exact setting, especially the data generating process, that is considered here.
- Choice of methods in Section 3.2: Since the authors perform a quick experiment on complete data to compare their architecture's performance to the results from random forest classifiers and regressors, why don't they also compare their method to random forests with missing values handling that does not require imputation (such as missing incorporated in attributes (Twala et al., 2008) that is implemented in the module sklearn.ensemble of scikit-learn or that can be implemented manually as suggested [here](https://rmisstastic.netlify.app/how-to/python/predict_html/how\%20to\%20predict))?
- Almost a minor comment: what is the impact of data standardization in the OpenML Data section (3.2), especially for the proposed method(s)?

### Minor comments (that did not impact the score)
- p.1: I would add that removing incomplete samples is even impossible in high-dimensional settings and not only an issue in small-sample size datasets.
- p.2: Some works exist in high-dimensional settings with missing values such as Jiang et al. (2021).
- p.2: anything in the rest of network $>>$ anything in the rest of _the_ network
- p.5: converge _slower_ $>>$ converge _more slowly_
- p.9: This feature is _clinical_ $>>$ This feature is _critical_?
- p.9: _several analytical and empirical_ aspects of PROMISSING _remains_ unexplored $>>$ _analytical and several empirical_ aspects of PROMISSING _remain_ unexplored (I didn't find any analytical aspects of PROMISSING discussed in the paper)

######################################
### Post-rebuttal update
I thank the authors for their detailed and timely responses. The proposed additional experiments on MAR and MNAR data support the authors' claims. I followed all exchanges between the authors and the other reviewers who criticize the lack of contributions/novelty of the proposed method. I agree to a certain extent with the other reviewers, especially concerning the justification and derivation of the proposed neutralizer as well as its theoretical grounding. However, I would acknowledge the authors' motivation to provide a simple-to-use method that allows to adapt the prediction indecisiveness to the level of available information which is in important feature, especially in medical contexts. In their extensive simulation study and real-world application they demonstrate the performance of this method and its comparison to other existing methods without claiming to perform better but being conceptually easier to integrate in existing architectures.
Therefore I will increase my score and recommend accepting this paper. In case of acceptance (or for a re-submission at another venue in case of rejection), I strongly encourage the authors to modify the introduction and the conclusion of their paper for the camera-ready to emphasize the current lack of theoretical understanding of this approach (currently they only write "some analytical and empirical aspects
of PROMISSING remain unexplored"). Pointing out this current lack of theoretical understanding won't hurt the contribution of this empirical work but could encourage other researchers to take up on these open questions and to try to theoretically understand the empirically observed behavior.

######################################

### References
[1] Wei Jiang, Małgorzata Bogdan, Julie Josse, Szymon Majewski, Błażej Miasojedow, Veronika Ročková, and TraumaBase® Group. Adaptive Bayesian SLOPE: Model Selection with Incomplete Data. _Journal of Computational and Graphical Statistics_, 2021.

[2] Marine Le Morvan, Julie Josse, Thomas Moreau, Erwan Scornet, and Gaël Varoquaux. NeuMiss networks: differentiable programming for supervised learning with missing values. _arXiv preprint arXiv:2007.01627_, 2020.

[3] Marine Le Morvan, Julie Josse, Erwan Scornet, and Gaël Varoquaux. What's a good imputation to predict with missing values?. _arXiv preprint arXiv:2106.00311_, 2021.

[4] Bheki ETH Twala, M. C. Jones, and David J. Hand. Good methods for coping with missing data in decision trees. _Pattern Recognition Letters_, 29(7), 2008.

**Summary Of The Paper:**

The present paper proposes an alternative to imputation or list-wise deletion in the context of neural networks and incomplete features. Missing values are replaced by a data-specific numerical representation that is learned at the same time as the rest of the network. The handling of the missing values is located in the neurons of the first layer and each missing value is "replaced" by a neuron-specific neutralizer in its activation function. They empirically show that their approach is comparable to several imputation techniques. In a clinical application they show that their model becomes indecisive as the amount of missingness (in terms of missing features) increases, which is a potentially interesting feature for clinical prediction models.

**Summary Of The Review:**

In summary, the proposal of this paper is interesting and promising, but it lacks clarification of the problem setting and justifications w.r.t. the choice of the representation of the missing values as well as the positioning of this approach w.r.t. other methods such as NeuMiss (Le Morvan et al., 2020). Importantly as well, a more extensive simulation study would be helpful to assess the behavior of the proposed strategy, especially looking at other missingness mechanisms than MCAR alone. I would encourage the authors to at least either add more justifications and/or results in the analytical part or to extend their simulation study to a wider range of possible settings to allow for an appropriate assessment of their proposal.
I will read the rebuttal carefully and am willing to increase the score if the authors address the raised concerns.

---

> ### Author Response · Authors · 2021-11-18
> **Additional experiments on MAR and MNAR will be added.**
>
> We would like to first thank the reviewer’s positive comments and for realizing the potentials of our contributions, the clarity of the text, and the reproducibility of our results. From the summary, it is clear that the reviewer has grasped the core messages in our paper. We have enjoyed reading the reviewer’s constructive comments and suggestions and we believe they are applicable and indeed will improve our work. The reviewer comments include 2 suggestions for complementary experiments (exploring MNAR setting, using ensemble approach), and 2 clarification comments (on the predictiveness of patterns of neutralizers and data standardization). Below we discuss in detail these suggestions and comments:
> -	We agree with the reviewer that the text may fail to effectively communicate the best-suited use cases of the proposed method. As suggested by reviewers, in the new version we will also include the results of our experiments for wider missing value mechanisms including MNAR. Our results on simulated data show that indeed the PROMISSING can learn the informative patterns of missingness in the MNAR setting. This is a crucial feature that puts PROMISSING on par with advanced and recent methods such as NeuMiss when applied to MNAR data. On the positive side, unlike NeuMiss, in PROMISSING 1) no imputation is needed (NeuMiss needs at least zero imputation), 2) no parameters are added to the model, i.e., there is no need for concatenating the inputs with an indicator vector (or mask) that doubles the parameter (thus complexity) of the neural network model. Based on our empirical results, we believe PROMISSING is the best approach for handling missing values with neural networks especially: 1) when we do not know the pattern of missingness in advance (thus it could be MCAR or MNAR), 2) for sensitive application of ML in an uncertain environment in which a wrong prediction has more consequences than a neutral prediction.
> -	As mentioned in the above response, to explore the hypothesis of the reviewer, we performed the simulation study on MNAR data and indeed the results are interesting. It seems indeed the learned patterns of neutralizer can be informative depending on the missing data mechanism. We will present the results in the new version modifying Fig.3 for the MNAR scenario (we needed to move this figure to the supplementary to save some space to accommodate reviewers' suggestions).
> -	The reviewer suggests adding a comparison with sklearn’s HistGradientBoosting to the experiments on OpenML data. We have included this comparison in the new version. Please bear in mind that this method cannot be used in our simulation study as it is a convergence study for NN-based models. Our results on the OpenML study shows that this method performs the best, which is another motivation to keep unknowns as unknown rather than using (in some case dangerous) data imputation. The PROMISSING method provides the same possibility for NN models.
> -	The reviewer asks for clarification on the effect of data standardization on results in section 3.2. The data in the OpenML datasets are very heterogeneous with variable ranges. Thus, using some rescaling approaches in advance is inevitable. Otherwise, as is expected, it will negatively affect the performance of some ML models including NNs regardless of missingness pattern (i.e., on complete data). When there is no missing value a PROMISSING network turns to an ordinary NN model thus data ranges affect its performance.  We agree that standardization is not always the best approach for rescaling the data, but we used it as a common practice in the field. Since it is used equally across different methods in our comparisons, it has no effect on the final conclusions.
> -	We have applied the reviewer’s minor suggestions to our best and enriched the bibliography with suggested studies.
>
> Thanks to the reviewer’s comments, we think the manuscript has improved in terms of experimental results and clarity of purpose. We will submit the revised version soon, meanwhile, we of course welcome the reviewer’s suggestions.

---

### Official Review · Reviewer_vNRV · 2021-11-03

**Correctness:** 2
**Technical Novelty And Significance:** 3
**Empirical Novelty And Significance:** 2
**Recommendation:** 3
**Confidence:** 5

**Main Review:**

Approaches that can readily for data with missing values are crucially important.

However, the approach here is based on handwaving, with intuitions that seem fragile. It does not empirically perform better than mean imputation.

The introduction claims "it has been shown that constant imputation is only effective when the missing features are not informative (Josse 2019)". This is not at all what Josse 2019 shows.

Page 3 claims that the approach enables relaxing the MAR assumption, but there is no legit argument to this claim.

The goal of neuralizing the effect of missing values on activations inside the architecture is not a desirable one. Indeed, suppose that we have two features, X1 abs X2, and y only depends on X2, but X1 and X2 are correlated. Should X2 be missing, an architecture attempting to make a prediction solely from the observed data and using the same logic add the fully-observed case, would fall, given that the fully-observed case relies only on X2. Rather, am adequate architecture would then rely on capturing and using the link between X1 and X2, and could achieve good predictions.

Figure 2 should show other approaches, such as mean imputation.

**Summary Of The Paper:**

This submission contributes an approach to handle missing values in Neural networks by replacing inside the architecture the missing values by placeholders which cancel the role of the feature in the architecture. The approach is benchmarked empirically, but does not appear to outperform mean imputation.

**Summary Of The Review:**

The contribution is based on intuitions that do not seem very solid and should be better studied. It does not really perform better than mean imputation.

---

> ### Author Response · Authors · 2021-11-18
> **The novelty and impact of our work is not in improving performance**
>
>
> We would like to first thank the reviewer for the comments and suggestions. The reviewer's comments suggest two text edits, two criticisms, and two questions. We believe the reviewer's concerns can be addressed by some text edits and more clarifications on the
> the motivation of our work.
> -	The reviewer believes our approach is only based on intuition. First, we do not see this as a negative point as all good ideas started with intuition. Second, we have found this comment quite unfair as we believe our ideas are supported with empirical evidence. Here, we present a principled approach to handle missing values in neural networks WITHOUT data imputation. Our basic PROMISSING approach does not add any extra parameter (in mPROMISSING it adds only 1) or hyperparameter to the model (unlike more complex imputation approaches such as KNN, MICE, or GAN-based methods that need to learn few-to-many extra parameters or hyperparameters). In several experiments on ~ 50 datasets, we empirically show that our method is 1) can reach the performance of a Bayes classifier on data with missing values, 2) learns meaningful representation for unknowns, 3) performs as well as baseline imputation methods, 4) is more reliable in uncertain environments, 5) can be used for counterfactual interpretation of complex ML models. We completely agree with the reviewer that this contribution can be augmented with extended theoretical analysis, but as stated in Sec. 4 and given the available space, we postpone this to our future work. To address the reviewer's comment, in the new version, we will be more explicit about the theoretical aspects of PROMISSING by framing its features in three propositions with detailed proof.
> -	The main criticism of the reviewer is around the argument that our method does not perform better than a mean imputer in section 3.2. First, we never claimed in the paper that PROMISSING provides a better (classification or regression) performance compared to other imputers. Instead, our focus in experiments in section 3.2, on several datasets from OpenML, is to show that by using PROMISSING (keeping unknown as unknowns), we do not lose any performance compared to its alternative imputation approaches. Our core argument in this paper is that adopting the PROMISSING approach is safer in more delicate applications of ML. Examples are autonomous driving, autopilot airplane systems, robotic surgery machines, or clinical decision-making. In these applications, it is common to miss the signal from some sensors (for example, due to sensor failure or occlusion). Do any of us (as ML method developers) dare to use (or even recommend) a mean imputer (or other imputers) to fill (guess?!) missing values in these applications? Our answer to this question is a big ‘NO’! We have tried to demonstrate an example of these delicate applications in our paper in psychosis prognosis prediction. We think it is nonsense and at the same time very dangerous, for example, to impute the missing blood tests results (cytokines) using demographics and behavioral measures. Even the best imputers in such situations are random guessers and may have catastrophic consequences for patients. Add to this the fact that many imputers do not work for categorical and non-ordinal feature types (for example, the mean imputer or even more complex GAN-based imputers), and many others need to tune extra hyperparameters (thus vulnerable to data distribution shift at the test time). This is while PROMISSING not only can deal with different feature types but also shows some promises for more reliable decision making when facing many unknowns (see results in Fig.6). To address the reviewer’s comment and to clearify our motivation, we can add a few sentences to the introduction to emphasize more the shortcomings of classical imputation approaches in more delicate applications. Furthermore, we would like to emphasize that, based on our experiments, the best strategy for handling missing values is always problem-specific, i.e., it is difficult to suggest a strategy that always works the best for every problem. Sometimes a simple imputer such as zero imputer works better than complex multiple imputer approaches (see also Jager et al. (2021)). The main feature that differentiates the PROMISSING approach from its alternatives is that it provides a means to learn the best imputation strategy from data using the ordinary backpropagation strategy. For example, as it is shown in Sec. 3.1 and Fig.3, the PROMISSING can learn to be a zero-imputer or mean imputer for some neurons. In the end, we believe that these strategies should not merely be judged based on their final performance, but we should also consider, for example, their computational complexity and reliability in decision making.

---

> ### Author Response · Authors · 2021-11-18
> **The comments, suggestions, and clarifications are minor and can be simply accommodated in the revised version.**
>
> -	We agree with the reviewer that the sentence “it has been shown that constant imputation is only effective when the missing features are not informative (Josse 2019)” is not accurate. We will revise it as “it has been shown that constant imputation is Bayes consistent when missing values are not informative (Josse 2019)”. We can also remove this sentence as it does not add much to our arguments in this paper. As pinpointed by the reviewer, our experiments also confirm the effectiveness of mean imputation (as an example of constant imputation) when the missingness is not informative (e.g., in the MCAR setting). We will enrich the new version of the paper with more analysis in MAR and MNAR settings.
> -	We agree that the following sentences can be misleading and convey a wrong message: “Furthermore, it is different from regression-based imputation and model-based approaches in the sense that a missing value in a specific feature is not inferred from other observed features, or the distribution of observed values, i.e., in PROMISSING unknowns remain unknown. Therefore, the MAR assumption in the multiple imputation and model-based approaches is relaxed.”  Here we intended to convey the message that: Unlike some regression and model-based approaches that assume (apriori) MAR on the missing data mechanism (thus fail if the missing data are not MAR), handling missing values in PROMISSING (neutralizing their effect on the activation of the neuron) does not assume any specific missingness mechanism (like constant imputation methods). This may or may not have consequences depending on the imputation strategy and covariance structure between observations (see Sec. 4 in Josse 2019). To address the reviewer’s comment, we will modify the sentence to avoid further confusion. Furthermore, in the new version, we will provide empirical evidence for the effectiveness of PROMISSING in the MNAR setting, in which the missingness patterns are informative.
> -	The reviewer believes using PROMISSING when there is a correlation between two covariates is suboptimal in terms of the performance of the model.  The example presented by the reviewer is similar to Example 2 page 13 Josse 2019, in which the conditional mean imputation will fail if there is no link between covariates. We believe the PROMISSING approach achieves its goal in reliable decision making in both scenarios when there is 1) a correlation between two covariates, and 2) when they are independent. To better understand the argument, let’s assume $x_1$ and $x_2$ represent the signal from two sensors used to control a robotic arm for a surgical operation. We assume $x_1$ and $x_2$ are partially correlated. Let the output $y$ represent the decision for whether to cut a vein or not, and we know that such a decision should be made only based on the signal from sensor $x_2$. In this scenario, if the $x_2$ is missing and, no matter whether the two covariates are partially correlated or not, the PROMISSING model becomes less decisive (when there is a correlation) or indecisive (when covariates are independent). We strongly believe that this cautious decision-making approach is more appropriate for sensitive applications. This is while if we use the mean imputation (or any other imputation approach), the model may come up with a very certain decision in a very uncertain environment. We certainly agree that it is worthwhile to analyze the behavior of PROMISSING for different covariance (or even causal) structures among observations. We are eager to invest some time answering these questions in the future (this is a full separate study like Josse 2019 in which the behaviors of constant imputers are analyzed).
> -	Figure 2 presents the results of our simulation study. As stated in the text, the goal of our simulation study is to analyze the behaviors PROMISSING approach on a simple toy data (not comparing the performance), including 1) its convergence behavior in comparison with the Bayes optimal classifier when the test set is with/without missing values, 2) examining the learned representations for unknowns. In the simulation study, we certainly did not aim to compare the performance of different imputation techniques (this is done in the OpenML experiments). To accommodate the reviewer's comment, we will include the performance of zero, mean, KNN, MICE, and other data imputation approaches in our simulation study.

---

> ### Author Response · Authors · 2021-11-18
> **Last words**
>
> In the end, we appreciate the reviewer’s comments, and we think the text after revision will be more clear and technically sound. Please bear in mind, improving the performance of the model is not the core contribution in this work and according to the ICLR review policy, the quality of submissions should not be judged merely on improving performance but novelty, relevance, impact, and added knowledge should also be considered (https://iclr.cc/Conferences/2022/ReviewerGuide).  We strongly believe in the novelty (this is the first method that can handle missing values in the neural network without any addition and only using internal mechanism and parameters of NNs) and impact (our contribution is one step ahead toward using ML models in high-risk applications) of our work, and we think it provides a simple yet principled approach to deal with missing values in NNs. We hope our adjustments and clarifications provide enough materials to the reviewer to evaluate our work more positively in the next stages. We will upload the revised version including adjustments and additional experiments in the next few days. Meanwhile, we are happy to take more suggestions and comments.

---

### Author Response · Authors · 2021-11-21
**General Rebuttal Letter**

We thank all reviewers for their effort and constructive suggestions. The revised manuscript is uploaded. We have already submitted a point-by-point response to all reviews, but we find it helpful to summarize our responses to major comments here. We have managed to apply almost all the reviewers’ comments and suggestions, and we are delighted with the outcome. Here is the summary of significant changes:
- We have included a set of three propositions in the method section (and their proof in the appendix) to clarify some theoretical aspects of the proposed method. In these propositions, we explain the motivation behind using neutralizers: “when all inputs to a PROMISSING neuron are missing, the activation of the neuron is zero.” Furthermore, we show that if there are no missing values in inputs, the PROMISSING neuron acts like a normal neuron, i.e., its activation is equal to a normal neuron. We have omitted these details in the first submission (they were only mentioned in the text), but it seems some reviewers came up with the misconception that our methodological contributions are all arbitrary and only based on intuition without any mathematical ground.
- Based on the reviewers’ suggestions, we have enriched our empirical evaluations with a set of new experiments including 1) experiments on MAR and MNAR on simulated data, 2) experiments on MAR and MNAR on OpenML data, 3) comparison with a more recent DL-based imputation method (GAIN), 4) comparison with other models (rather than NNs) that can directly handle the NaNs in inputs (Histogram-based Gradient Boosting). In these experiments, we had exciting observations, especially when applying PROMISSING to MNAR data. However, our general conclusions remained unchanged: when receiving incomplete data, the PROMISSING performs as well as models trained on imputed data; without the need for imputation or adding extra parameters or hyperparameters to the model.
- We have also tried our best (given the page limitation) to incorporate all the reviewers’ minor comments by adding more references, clarifications, and modifications.

**We argue that our method is novel and useful:**

- Novelty 1: This is the first method for directly dealing with missing values (without data imputation) in neural networks without adding extra parameters and hyperparameters to the network and only using its weights and biases. Furthermore, it relaxes the need for imputing incomplete data without compensating the model performance. We strongly believe this is an impactful addition to neural network theory, as it can be easily adopted, implemented, and applied in all different kinds of neural network layers (including dense, CNN, LSTM, etc.).
- Novelty 2: PROMISSING provides a genuine way for incorporating the uncertainty of the environment (defined by the lack of sensory inputs) in model decision making where the more are unknowns the less decisive is the model. This can have a significant impact on more delicate applications of ML in real-world scenarios, for example in autonomous driving or clinical decision making.

We have provided extensive empirical results to support our claims on ~50 datasets and in different missing value scenarios including MCAR, MAR, and MNAR. One of the main criticisms of our work is the performance of the models. In the end, we should reemphasize that we never claim PROMISSING improves the prediction performance of the models (in fact, why a model that receives less information should be more accurate) we claim that feeding NaNs to your PROMISSING model does not have a negative effect on its performance.
We again thank the reviewers for their effort and we are open to further discussing these points during the discussion phase.

---

### Author Response · Authors · 2021-12-02
**Summarizing the second round discussion**

We thank again the reviewers for engaging in the second phase discussion. Hereby, we would like to summarize the moments of the discussion:

- We appreciate DWUQ's feedback on the revised version. The reviewer commends the additional experiments added to the revised version and realizes the potentials of the proposed method in medical applications. The reviewer recommends accepting our contribution in the post rebuttal comment (we may be mistaken but this is not reflected in the final recommendation). The reviewer, however, has some concerns regarding the lack of theoretical grounds of the proposed method and suggests emphasizing this in the text. We agree with the reviewer that more extensive experiments are needed to fully understand all the theoretical and experimental corners of the proposed method (considered as future work), thus, to address the reviewer's concern we will modify the text (abstract and summary) accordingly.

- vNRV, surprisingly, released his/her post-rebuttal recommendation before reading the revised version (on Nov. 19 while we uploaded the revised version on the 21st). We did not find this a good practice at all. The reviewer vaguely criticizes the so-called "light contribution" of our work without any practical suggestion on how we can improve our work. He/She refused to engage in the discussion, thus, we did not find any opportunity to resolve/clarify possible misunderstandings.

-  We have received very positive feedback from Bfyk about the revised version stating, "It is much improved in my opinion.". We really appreciate the reviewer's engagement in the discussion, but we just wonder why this positive evaluation is not reflected in the final rating. The reviewer has some suggestions on including more baselines (GRAPE) and the choice of performance measures (AUC drop vs. AUC). To address the reviewer comments, we will include the results for the GRAPE in the main text and the AUC-based evaluations in the supplementary (the added results have no effect on the conclusions of our work).

- Despite our extensive response, reviewer FmEA refused to engage in the discussion. The reviewer's statement "I vote for accepting the paper subject to some additional experiments, based on my review above" lighted up flames hope and enthusiasm in us for running a very extensive set of experiments in a short time, and receiving zero feedback is hugely demotivating. However, we still appreciate the constructive comments after the first phase. They improved our work significantly.

- The exhaustive discussion with reviewer eatM was joyful and at the same time frustrating (we should apologize for some burst moments in our response. We have no hard feelings and are open to possible collaboration after this review process). The reviewer questions every aspect of our method (it is no more than zero imputation, it can be implemented using an indicator vector, it is all based on intuition, it is already in the previous work) to justify his/her unfair judgments of the novelty of our work. We have provided a very detailed set of responses to maybe convince the reviewer otherwise but we apparently failed. We also willingly agreed to modify the text accordingly to satisfy the reviewer's main (in our opinion very minor) comments including 1) explaining the difference between zero imputation and PROMISSING in the main text,  2) adding the results for zero imputation to Fig. 4, 3) including the results for the indecisiveness of PROMISSING models on full OpenML data in the supplementary. Receiving no positive response from the reviewer despite our positive attitude to address the reviewer's comments leaves us with the only conclusion that his/her judgments may have non-scientific and unconstructive motivation.

In the end, we of course cannot deny the majority of constructive effects of this review process on our work. We have learned a lot and the manuscript is improved substantially. Moreover, we cannot claim the manuscript is perfect. However, we think some positive aspects of our work are undervalued (a straightforward method to deal with nans in NNs without extra complexity, the first-ever method to learn a representation for unknowns in ML, awareness about the problem of decision making in uncertain environments and proposing a possible solution to that) while its (in our opinion minor) shortcomings are overly bolded. We still think this contribution could provide a ground for a further stimulating discussion on the important problem of missing values at ICLR.

Thank you all again,
Sincerely,
The authors of Paper2697

---

### Decision · Program_Chairs · 2022-01-20

**Decision:**

Reject

**Comment:**

The paper presents a simple and intuitive method to prune the missing value in the learning and inference steps of the neural networks, leading to similar prediction performance as other methods to impute missing value. It has some really useful insights, but could benefit from one more round of revision for a strong publication:
1. improving the writing so that its sets up the right expectations on the contributions of the paper;
2. providing discussions on its connections (and differences) with zero-imputation and missing-indicator methods;
3. thoroughly investigating the experiment results to illustrate the advantages of the proposed method.

The recommendation of reject is made based on the technical aspect of the paper.
-----------------------------
During the rebuttal phase, the authors misused the interactive and transparent (for the better or worse) openreview system by writing inappropriate comments with personal accusations to the reviewers who write negative reviews. We would like to extend the apologies to the reviewers for this unpleasant experience and thank the reviewers for their engagement and work, as well as their fair assessment of the paper.